# Exact Methods and Heuristics for Order Acceptance Scheduling Problem under Time-of-Use Costs and Carbon Emissions

**Mariam Bouzid** *[ID], **Oussama Masmoudi** [ID] and **Alice Yalaoui** [ID]

Computer Laboratory and Digital Society (LIST3N), University of Technology of Troyes, 10000 Troyes, France; oussama.masmoudi@utt.fr (O.M.); alice.yalaoui@utt.fr (A.Y.)
* Correspondence: mariam.bouzid@utt.fr; Tel.: +33-2571-8089

**Abstract:** This research focuses on an Order Acceptance Scheduling (OAS) problem on a single machine under time-of-use (TOU) tariffs and taxed carbon emissions periods with the objective to maximize total profit minus tardiness penalties and environmental costs. Due to the NP-hardness of the considered problem especially in presence of sequence-dependent setup-times, two fix-and-relax (FR) heuristics based on different time-indexed (TI) formulations are proposed. A metaheuristic based on the Dynamic Island Model (DIM) framework is also employed to tackle this optimization problem. These approached methods show promising results both in terms of solution quality and solving time compared to state-of-the-art exact solving approaches.

**Keywords:** single machine scheduling; order acceptance; green scheduling; time-indexed; fix-and-relax

## 1. Introduction

In this period of economic recession stemming from the COVID-19 pandemic [1] and coupled with the climate emergency, the implementation of effective policies and tools remains crucial to tackle current challenges. According to the International Energy Agency, the industrial sector accounted for almost 28% of the energy use in 2018, whereas the current crisis is likely to shift the industrial output to more energy-intensive manufacturers [2]. Consequently, the International Panel on Climate Change urges governments and economic actors to engage rational and coordinated responses to the climate change through a sustainable development. The latter sets on economic, social and environmental pillars guarantying prosperity, social justice and nature conservation. In addition to scientific concerns, civil society is increasingly calling for sustainable development, pressuring governments and companies to adhere to ethical standards and green framework.

To curb GHG (greenhouse gas) emissions, for instance, the implementation of European Union Emission Trading System (EU ETS) for energy-intensive industry has been a keystone of EU energy policy. EU ETS forms a 'cap and trade' scheme allowing the companies to emit and exchange GHG allowances, decreasing on yearly basis. Heavy fines are applied if the allowance is not complied. Since its introduction, it has achieved a 8.7% cut on European GHG emissions [3]. To complement EU ETS, members states introduce taxes on carbon whenever emissions exceeds a given threshold. In this context, this paper especially focuses on taxes on $CO_2$ emissions. Along with these regulations, monitoring energy consumption and carrying out an energy audit is now mandatory for companies that meet specific criteria. In our economy based on supply and demand, the industrial sector has been establishing itself as a major actor. For a manufacturing company, the production of goods to satisfy customers demand generates profit, investment and employment. Altogether, this participates in the real economy as the manufacturing sector contributes to nearly 20% of global gross domestic product [4]. Therefore, the sector must adapt as effectively as possible to the newest regulations while maintaining competitiveness. This past decade, the environmental impact of the supply chain has been widely studied, suggesting that the operational level is more prone to profound changes. Indeed, integrating

energy aspects into planning and scheduling, besides replacing obsolete equipment, is one of the most cost-effective ways to attain sustainability objectives [5] such as reducing GHG emissions and energy consumption. However, a trade-off between reducing energy consumption and productivity is always noticeable [6]. Nevertheless, energy sobriety is beneficial for both economic and environmental reasons. First, energy cost is a shortfall for heavy energy-using industry as the energy supplies are becoming expensive. For this purpose, smart grid technologies have already been deployed. In this context, energy providers have designed preferential tariffs rate such as time-of-use (TOU), real-time or critical peak pricing. TOU rates incite manufacturers to shift their production to cheaper off-peak hours instead of on-peak hours. Second, depending on the energy mix used (e.g., coal or gas based), reducing energy consumption or costs is a direct way to reduce GHG emissions [7].

Early industry focused on mass production with high volumes of few products. Yet, this past decade, major changes in industry have been occurring [8]. Product variety and demand for tailor-made products force manufacturers to make a compromise between their available production capacity, their production organization and their sales volume. This global trend appears in a plethora of manufacturing sectors from textiles to the food industry. Indeed, this can be observed in real manufacturing settings, such as printing processes in packaging companies [9], service delivery including food or critical equipment maintenance companies facing time windows, high demands with limited resources deployment [10,11]. Order acceptance scheduling (OAS) is an abstraction to model this particular trend. In this vein, this paper investigates a single machine OAS problem with release date and sequence-dependent setup times under TOU tariffs and taxed carbon emissions. In this problem, the company has to decide which order to produce, among $n$, and establish a schedule accordingly. Each order is available within a specific time window. Moreover, a setup operation is performed between orders, and its duration depends on the previous order sequenced. The objective is to maximize the total revenue of orders minus tardiness penalties while meeting clients deadlines and green manufacturing considerations. The latter has an impact on the orders selection and their schedule. Chen et al. [12] are the first to introduce this problem while proposing a disjunctive Mixed Integer Linear Program (MILP). Bouzid et al. [13] consider an arc-time-indexed (ATI) MILP to cope with the high complexity of this NP-hard problem and successfully solve some large instances. However, these approaches are limited by design and thus require the use of heuristics.

For this purpose, this paper analyses two time-indexed (TI) formulations and two Fix-and-Relax (FR) heuristics applied to the provided formulations. Moreover, an island-based genetic algorithm (GA) first proposed by Candan et al. [14] is developed . The contributions of this paper are threefold. First, we refine mathematical models on this problem. Second, we give an overview of the formulations behavior with respect to the FR heuristic , and finally we propose a competitive and robust metaheuristic to fill the gap in the literature and improve existing results. Section 2 presents a literature review on OAS under energy aspects and more globally on scheduling considering energy. Section 3 presents in details the considered problem. Section 4 is dedicated to the mathematical formulations, and Section 5 presents the solving methods. Benchmark and experimental design are introduced in Section 6 along with the results and their interpretation. Conclusion are drawn and perspectives are given in Section 7.

## 2. Related Literature

This section presents existing literature on scheduling and OAS problems under energy aspects with their developed solution approaches. First, an overview on scheduling problems incorporating environmental aspects is given. Next, related works on OAS problems are introduced. Before concluding, a focus is made on the resolution techniques.

Gao et al.'s [15] review on scheduling problems under energy aspects reveals that this topic has been growing in interest in recent years. Complex shop systems, including job shop and flow shop, represent the majority of the studies, whereas single machine

features less than 4% of their corpus. One of the most important points highlighted in this study is that energy efficiency is conceptualized by two approaches. First and foremost, it is done by introducing it as a criterion. Indeed, targeted criteria such as Total Energy Consumption or Costs (TEC) or total carbon emissions have been successfully incorporated into scheduling problems in numerous work [16–19]. Second, energy efficiency is modeled by dedicated constraints coupled with a classical scheduling objective [20,21]. For example, in the work of Liao et al. [20], weighted tardiness and completion times are minimized while satisfying a periodic threshold on energy consumption for a single machine.

In the literature, different assumptions relative to the energy aspects may be encountered. These assumptions are related to the quantity considered (carbon emissions, energy consumption, power etc.), the machine characteristics (energy states, speed), the variation of the energy costs during the day or the system involved (single machine or shop systems). Depending on these assumptions, the problem entails particular properties and thus is solved with specific approaches.

For single machines, Mouzon and Yildirim [5] present an adaptive search metaheuristic to minimize TEC and total tardiness. In their study, they examine the idle, setup and processing energy of the machine in order to efficiently adjust the production and avoid tardy jobs. The neighborhood move developed by the authors inserts setup or idle times between jobs, which can reduce energy costs but can lead to tardiness. Che et al. [22] consider a TI MILP to minimize TEC under TOU electricity tariffs and develop a greedy insertion heuristic which moves jobs to the off-peak periods. Aghelinejad et al. [23] propose a dynamic program for the single machine problem under TOU tariffs considering machine states and investigate the complexity of various energy costs strategies that can induce the problem to be polynomial.

As for shop scheduling or parallel machines, other energy aspects are studied. Zhang et al. [24] address a speed scaling job shop problem with the objective to minimize both tardiness and TEC. In their work, they monitor machines speed to efficiently modulate the production process. Dedicated local search procedures are designed to cope with the complexity of the problem. In the same vein, Jiang et al. [25] consider energy consumption per time unit and idle energy consumption minimizing TEC and makespan. They employ an Evolutionary Algorithm (EA) in their solving approach. In [26], the authors optimize the TEC and the makespan of unrelated parallel machines under time-and-machine-dependent electricity costs. Their solving approach involves an hybrid GA that incorporate an idle-time insertion procedure to cut costs on electricity expenses. Considering $CO_2$ emissions, Foumani and Smith-Miles [27] assess common carbon reduction policies on a flow shop. One of their conclusion confirms that optimizing the schedule plays a key role in the reduction of $CO_2$ emissions rather than changing equipment. In the meantime, they show that the 'cap and trade' approach is a cost-effective policy. In [7], a flowshop under time-dependent electricity tariffs and $CO_2$ emissions is tackled. The authors propose a TI MILP to minimize simultaneously carbon footprint and TEC with machines having different consumption levels. Their study suggests that a trade-off between electricity costs and $CO_2$ emissions appears when the energy providers are coal-based. As in [7,12], the assumption on time-dependent $CO_2$ emissions and electricity costs holds for this paper.

OAS is a particular scheduling problem where the decision covers the selection of a subset of orders, among $n$, and their sequencing in a capacity-constrained production system. Typically, this entails a fixed time frame to complete orders and an associated cost-driven event where the company fails to produce within the time-window. The solution space of OAS problem extends classical scheduling one, as jobs can be accepted or not. Indeed, at worst the number of possible solutions in OAS problem is $\sum_{k=1}^{n} k!$, where all the $k$-permutations of $n$ without repetition are considered, whereas only $n!$ solutions form the solution space in classical scheduling problems. In the literature, for both single- or multi-machine systems, a variety of configurations of problems are considered such as sequence-dependent setup times [9,28,29], preemption [11] or resource constraints [10,30]. As for our

research, the immediate related works are those presented in [9,28,29]. A comprehensive survey [31] presents an overview on OAS problems, while in [32], a focus is made on scheduling problem with rejection.

OAS involves mainly economic-related criteria, primarily embodied by the maximization of the total profit generated by the orders. Service level [33], percentage of accepted orders or utilization can also be maximized in OAS problems. In addition, cost penalties can be integrated in the objective function when tardiness or order rejection occur. For instance, Oguz et al. [9] maximize the total profit of accepted orders minus their possible tardiness penalties. As in scheduling problem, OAS solving methods involve exact and heuristic approaches. MILP [9,28,29], dynamic programming [34] and branching methods [35] have been employed for various OAS problems. In the meantime, as these problems are mostly NP-hard, a wide range of metaheuristics from local search to EA have been utilized and have shown very robust performances [36–38]. Besides, reports in the literature describe an order-based and a time-based FR heuristics applied to an OAS problem under resource constraints [30] that both achieve a tighter gap for large instances. A recent work of Tarhan et Oğuz [39] proposes a two-phase matheuristic that exploit a time-indexed model. First, they assign orders to time segments using the relaxed version of their model and generate a schedule subsequently; the solution is then improved by a VNS. This process is repeated until the termination criterion is met. However, energy aspects are not considered in their work.

Literature on OAS considering resource constraints and/or energy aspects is very sparse. Garcia [10] tackles an resource-constrained OAS problem with the objective to maximize profit with rejection penalties using an EA and a priority rule heuristic. Kong et al. [40] maximize the net revenue of a parallel machines system with order acceptance and a global constraint on the energy consumed by machines and their launch budget. In their work, a comparative analysis between diverse variable neighbor search algorithms and a dynamic programming approach is conducted. Considering electricity tariffs, to the best of our knowledge, three papers have been reported [12,13,41]. These papers follow up the works in [12,13] that both investigate the OAS problem under TOU and $CO_2$ emissions periods and sequence-dependent setup times with a disjunctive formulation in [12] and an ATI model in [13]. Moreover, this paper contributes to the comprehension of the OAS under energy aspects by introducing approached solving methods that improve the existing results.

A vast majority of the investigated problems on scheduling are NP-hard or pseudo-polynomial, justifying the significant use of heuristics that can outperform exact methods. Likewise, matheuristics are developed to tackle the aforementioned problems.

FR heuristic is a model-based heuristic which is applied in planning and scheduling problems. Promoted by Absi et al. [42], this heuristic has been employed in production planning researches considering energy aspects [43,44]. For instance, Masmoudi et al. [43] minimize TEC for a single-item capacitated lot sizing problem in a flow shop with TOU tariffs and power constraints. Their FR strategy relies on the relaxation of the binary decision variables involved in the time-dependent constraints. Besides, the FR heuristic is also used for scheduling problems such as operating rooms scheduling [45,46] and harvest scheduling [47]. In [45], Silva et al. maximize the use of operating rooms with constraints on the staff schedule and skills. Following the advances of Industry 4.0, a recent study of Li et al. [48] features a GA combined with machine learning approaches that minimize makespan for a job shop rescheduling production system. The machine learning techniques aim at evaluating rescheduling patterns. Their framework is able to outperform classical approaches with less configuration changes made at the right time. The survey of Dolgui et al. [49] summarizes the contours of scheduling problems from the point of view of optimal control. In the context of complex systems, this approach appears to answer to the new challenges raised by the Industry 4.0. Finally, Q-learning techniques have also been employed in [50] for an online single machine scheduling with the objective to minimize tardiness in the context of a smart factory [51]. This research compares the performances

of classical scheduling methods against reinforcement learning techniques and concludes that the latter can improve the resolution quality and time [25,52–54] . In this vein, the island-based framework introduced in [14] is a good candidate for solving combinatorial optimization problems such as scheduling. This framework proposes a self-adaptive migration policy between islands of individuals to efficiently explore and intensify the search. As in reinforcement learning techniques, the utility of the mutation operators, which control the number of individuals in each island, are re-evaluated at each iteration depending on the past performances. Good results have been presented for the 0/1 knapsack problem and the MAX-SAT problem in [55]. Therefore, this paper proposes an island-based metaheuristic as well as two FR heuristics based on two distinct exact models in order to efficiently solve the OAS problem with released dates and sequence-dependent setup times under TOU and taxed $CO_2$ emissions.

To finish this section some conclusions can be drawn. First, with the growing interest on environmental issues, both industrial and academics are paying more attention to incorporate them in their production and their models. Second, in the current economic climate, OAS problems find numerous applications; this is due to their capacity to introduce constraints on resources that usually are assumed unlimited. Last, metaheuristics, or more globally, artificial intelligence approaches, are privileged over exact methods. Moreover, the current trend is to use novel machine learning techniques as standalone solving approaches or to boost heuristics.

## 3. Problem Description

The OAS with sequence-dependent setup times, release date under TOU costs and taxed carbon periods is investigated in this paper. Parameters and notations are detailed in this section (Table 1).

**Table 1.** Parameters nomenclature.

| Symbol | Description | Units |
|:---:|:---:|:---:|
| $n$ | Number of orders | |
| $T$ | Planning horizon | min |
| $p_j$ | Processing time of order $j = 0, \ldots, n$ | min |
| $r_j$ | Release date of order $j = 0, \ldots, n$ | min |
| $d_j$ | Due date of order $j = 0, \ldots, n$ | min |
| $\bar{d}_j$ | Deadline of order $j = 0, \ldots, n$ | min |
| $e_j$ | Revenue generated by order $j = 0, \ldots, n$ | $ |
| $\Omega_j$ | Power consumption of order $j = 1, \ldots, n$ | kWh |
| $w_j$ | Tardiness penalties of order $j = 0, \ldots, n$ | min |
| $s_{ij}$ | Setup-times between order $i = 0, \ldots, n$ and order $j = 1, \ldots, n$ | min |
| $m$ | Number of TOU intervals | |
| $b_k$ | Starting time of TOU interval $k = 1, \ldots, m$ | min |
| $EC_k$ | Electricity cost of TOU interval $k = 1, \ldots, m$ | $/kWh |
| $h$ | Number of $CO_2$ emission intervals | |
| $g_l$ | Starting time of $CO_2$ interval $l = 1, \ldots, h$ | min |
| $q_l$ | Amount of $CO_2$ emission per kWh in interval $l = 1, \ldots, h$ | kg/kWh |
| $Tax$ | Tax per kg of $CO_2$ emitted | $/kg |
| $c_{jt}$ | Energy cost of order $j = 1, \ldots, n$ at time $t = r_j, \ldots, \bar{d}_j$ | $ |
| $f_{jt}$ | Profit of order $j = 1, \ldots, n$ at time $t = r_j, \ldots, \bar{d}_j$ | $ |

Each order $j = 1, \ldots, n$ is completely defined by its processing time $p_j$, release date $r_j$, due date $d_j$, deadline $\bar{d}_j$, revenue $e_j$, power consumption $\Omega_j$ and tardiness penalties $w_j$. In addition, a sequence-dependent setup time $s_{ij}$ is defined between any pair of orders $i$ and $j$. A dummy order 0 is introduced in order to start the sequence. Each of its properties are set to zero except its setup time $s_{0j}$ between any order $j$.

An order $j$ is accepted when it is sequenced in the span ranging from its release date $r_j$ to its deadline $\bar{d}_j$ and rejected otherwise. A tardiness penalty $w_j$ is subtracted to an order

revenue $e_j$ for each time unit beyond its due date $d_j$. In Figure 1, $w_j$ represents the slope of the revenue decay between $d_j$ and $\bar{d}_j$. Moreover, in the original work, the planning horizon is divided into intervals with fluctuating TOU tariffs and $CO_2$ emissions. Each TOU interval $k = 1, \ldots, m$ is characterized by a starting time $b_k$ and an electricity cost $EC_k$. Each $CO_2$ emissions interval $l = 1, \ldots, h$ is determined by a starting time $g_l$ and an amount of $CO_2$ per kg and a *Tax* per emitted kg of carbon. As in [7,12], by assumption, $CO_2$ emissions are time-dependent, i.e., the emitted amount fluctuates over the day as the employed power sources are coal-based during off-peak hours and gas-based during mid-peak and on-peak hours.

Revenue

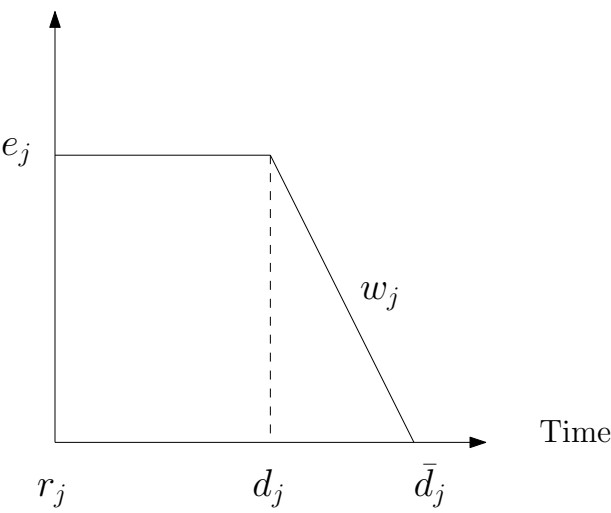

**Figure 1.** Profit calculation of an order $j$ based on tardiness (reprinted from [12]).

In this problem, the objective is to maximize the revenue minus tardiness penalties and energy costs. For simplification reasons, the energy costs can be calculated at each time-slots rather than at each intervals, especially as TOU and $CO_2$ emissions intervals partition differently the horizon. The energy cost $c_{jt}$ for any order $j = 1, \ldots, n$ at any period $t = r_j, \ldots, \bar{d}_j$ is thus computed with the formula given in Equation (1).

$$c_{jt} = \frac{\Omega_j}{60} \left( \sum_{k=1}^{m} EC_k \mathbb{1}_{b_{k-1} \leq t < b_k} + Tax \sum_{l=1}^{h} q_l \mathbb{1}_{g_{l-1} \leq t < g_l} \right) \tag{1}$$

The energy cost of each time period $t$ and for each order $j$ corresponds to the sum of the respective TOU and $CO_2$ taxed emissions costs of the examined period $t$ multiplied by the order's energy consumption expressed into minutes. In this expression, the indicator function $\mathbb{1}_x$ takes value 1 if condition $x$ holds, and 0 otherwise. In addition, some assumptions are stated in this problem. Preemption is not allowed, idle time energy is negligible . Setup and production use the same amount of energy. The planning horizon ends at the maximum of deadlines, that is, $T = \max\limits_{j=1,\ldots,n} \bar{d}_j + 1$.

## 4. Exact Approaches

The initial solving approach for this problem involves a sequence-based or disjunctive MILP proposed by Chen et al. [12]. Their model is based on integer decision variables that represent starting times, completion times and tardiness of each order, whereas the sequence is determined by binary decision variables defined between each pair of orders. Acceptation of orders are handled by binary decision variables. Due to its inherent properties, the disjunctive MILP is not as efficient for some instances with particular features. An ATI model is developed in [13] that can overcome some aspects of the disjunctive model.

In this paper, two new TI formulations deriving from the work in [41] that explored the problem without sequence-dependent setup times are presented and achieve better results than the disjunctive formulation. In Section 4.1, an On/Off formulation is presented. The model in Section 4.2, referred as TI Pulse, is an equivalent model for the same problem. Performances comparison between each MILP is presented in the last subsection.

### 4.1. On/Off Formulation

In this model, each binary decision variable $x_{jt} = 1$ indicates whether the order $j$ is processed at time $t = r_j, \ldots, \bar{d}_j$, or not $x_{jt} = 0$. In the same way, the binary decision variables $y_{jt} = 1$ corresponds to a unit of processed setup of an order $j = 1, \ldots, n$ at time $t = r_j, \ldots, \bar{d}_j - p_j$. For any order $j = 0, \ldots, n$, the binary decision variable $a_j$ takes value 1 if order $j$ is accepted; 0 otherwise. For any pair of orders $i, j = 0, \ldots, n$ the binary decision variable $u_{ij}$ equals 1 if order $i$ precedes directly order $j$. Finally the integer decision variables $T_j \in \mathbb{N}$ represent the tardiness of any order $j = 1, \ldots, n$ and $C_j \in \mathbb{N}$ its completion time. The MILP for the TI On/Off formulation is written as follows.

$$\textbf{maximize} \sum_{j=1}^{n} \left( a_j e_j - w_j T_j - \sum_{t=r_j}^{\bar{d}_j} (x_{jt} + y_{jt}) c_{jt} \right) \tag{2}$$

$$\sum_{j=1}^{n} (x_{jt} + y_{jt}) \leq 1 \qquad t = 1, \ldots, T \tag{3}$$

$$C_j \geq (t+1)(x_{jt} - x_{jt+1})$$
$$j = 1, \ldots, n \quad t = r_j, \ldots, \bar{d}_j \tag{4}$$

$$C_j \leq \bar{d}_j a_j \qquad j = 1, \ldots, n \tag{5}$$

$$T_j \geq C_j - d_j a_j \qquad j = 1, \ldots, n \tag{6}$$

$$\sum_{\substack{j=1 \\ i \neq j}}^{n} u_{ij} \leq a_i \qquad i = 0, \ldots, n \tag{7}$$

$$\sum_{\substack{j=0 \\ i \neq j}}^{n} u_{ji} = a_i \qquad i = 1, \ldots, n \tag{8}$$

$$\sum_{j=1}^{n} x_{jt} = p_j a_j \qquad j = 1, \ldots, n \tag{9}$$

$$\sum_{t'=r_j}^{t-p_j} x_{jt} + \sum_{t'=t+p_j}^{\bar{d}_j} x_{jt} \leq p_j (1 - x_{jt})$$
$$j = 1, \ldots, n, \quad t = r_j, \ldots, \bar{d}_j \tag{10}$$

$$\sum_{t=r_j}^{\bar{d}_j - p_j} y_{jt} \leq \sum_{\substack{i=0 \\ i \neq j}}^{n} s_{ij} u_{ij} \qquad j = 1, \ldots, n \tag{11}$$

$$y_{jt} - y_{jt+1} - x_{jt+1} \leq 0$$
$$j = 1, \ldots, n \quad t = r_j, \ldots, \bar{d}_j \tag{12}$$

$$\sum_{t'=r_i+1}^{t-1} x_{it'} \geq p_i (u_{ij} + y_{jt} - 1)$$
$$i = 0, \ldots, n \quad j = 1, \ldots, n \quad i \neq j \quad t = r_j, \ldots, \bar{d}_j \tag{13}$$

$$\sum_{t'=r_j}^{t-1} y_{jt'} \geq s_{ij}(u_{ij} + x_{jt} - 1) \tag{14}$$

$$i = 0, \ldots, n \quad j = 1, \ldots, n \quad i \neq j \quad t = r_j, \ldots, \bar{d}_j$$

$$\sum_{t=0}^{r_j-1} x_{jt} = 0 \qquad j = 1, \ldots, n \tag{15}$$

$$\sum_{t=0}^{r_j-1} y_{jt} = 0 \qquad j = 1, \ldots, n \tag{16}$$

$$\sum_{t=\bar{d}_j+1}^{T} x_{jt} = 0 \qquad j = 1, \ldots, n \tag{17}$$

$$\sum_{t=\bar{d}_j-p_j+1}^{T} y_{jt} = 0 \qquad j = 1, \ldots, n \tag{18}$$

$$x_{00} = 1 \tag{19}$$

The objective function (2) is the maximization of the total profit, i.e., the revenue minus the possible tardiness penalties and the environmental costs. Constraints (3) state that at each time the machine is either doing nothing, processing an order or doing a setup operation. Constraints (4) compute the completion times of order $j$ by retrieving the instant $t + 1$ when the production ends, that is, when $x_{jt} = 1$ and $x_{jt+1} = 0$. Constraints (5) limit the completion time of an accepted order $j$ to its deadline. Constraints (6) refer to the calculation of the tardiness of an accepted order $j$ with its completion time minus its due date. Constraints (7) indicate that an accepted order can have at most a successor. Constraints (8) impose that each accepted order must have a predecessor. Constraints (9) impose to any order $j$ to be processed exactly $p_j$ time units. Constraints (10) guarantee non-preemption by forcing the contiguity of the decision variables $x_{jt}$. To be more specific, if at time period $t$ order $j$ is produced ($x_{jt} = 1$), constraints exclude production $p_j$ units before and after $t$ in $r_j, \ldots, t - p_j$ and $t + p_j, \ldots, \bar{d}_j$. Implicitly, this means that order $j$ is produced in the interval $t - p_j + 1, \ldots, t + p_j - 1$. Constraints (11) define the setup operation of at most $s_{ij}$ time units between each order $j$ and its predecessor order $i$. Constraints (12) determine the continuity of the setup operation while guarantying that it should be done right before the processing of an order. Meaning that if at time $t$, order $j$ is in setup ($y_{jt} = 1$), either the setup operation is carried out ($y_{jt+1} = 1$) or the production starts ($x_{jt+1} = 1$). Constraints (13) establish the precedence relationship between a predecessor order $i$ and the sequence-dependent setup operation of order $j$. More precisely, if at period $t$, order $j$ is in setup ($y_{jt} = 1$) and order $i$ precedes order $j$ ($u_{ij} = 1$), the order $i$ must be completely processed before. This means that order $i$ is produced at least $p_i$ time units from $r_i + 1$ to $t - 1$, otherwise, the right hand-side is canceled. Constraints (14) establish the precedence relationship between the processing of an order $j$ and its sequence-dependent setup operation $s_{ij}$. It means that if order $j$ is in production at period $t$, and order $i$ precedes order $j$, the order $j$ should have been setup before, during $s_{ij}$ time units, from $r_j$ to $t - 1$. Constraints (15)–(18) ensure that each order cannot be processed or setup before its release date and after its deadline. Finally, constraint (19) forces the dummy order to start the sequence at time 0.

### 4.2. Pulse Formulation

The decision variables $z_{jt}$ in the Pulse model refer to the possible instants $t = r_j, \ldots, \bar{d}_j - p_j + 1$ when the order $j = 0, \ldots, n$ starts. It means that $z_{jt} = 1$ if and only if order $j$ starts its production at time period $t$, and 0 otherwise. Finally, for each pair of orders $i, j = 0, \ldots, n$ the binary decision variable $u_{ij}$ equals 1 if order $i$ precedes directly order $j$. In this formula-

tion, the $f_{jt}$ term represents the profit of an order $j = 1, \ldots, n$ at period $t = r_j, \ldots, \bar{d}_j - p_j + 1$ minus the tardiness penalties: $f_{jt} = e_j - w_j \max(0, t - d_j)$.

$$\textbf{maximize} \sum_{j=1}^{n} \left( \sum_{t=r_j}^{\bar{d}_j - p_j + 1} z_{jt} \left( f_{jt+p_j-1} - \sum_{t'=0}^{p_j-1} c_{jt+t'} - \sum_{\substack{i=0 \\ i \neq j}}^{n} u_{ij} \sum_{t'=1}^{s_{ij}} c_{jt-t'} \right) \right) \tag{20}$$

$$\sum_{j=1}^{n} z_{jt} \leq 1 \qquad t = 1, \ldots, T \tag{21}$$

$$\sum_{\substack{j=1 \\ i \neq j}}^{n} u_{ij} \leq \sum_{t=r_i}^{\bar{d}_i - p_i + 1} z_{it} \qquad i = 0, \ldots, n \tag{22}$$

$$\sum_{\substack{j=0 \\ i \neq j}}^{n} u_{ji} = \sum_{t=r_i}^{\bar{d}_i - p_i + 1} z_{it} \qquad i = 1, \ldots, n \tag{23}$$

$$\sum_{t=r_j}^{\bar{d}_j - p_j + 1} z_{jt} \leq 1 \qquad j = 1, \ldots, n \tag{24}$$

$$\sum_{t=r_i}^{\bar{d}_i - p_i + 1} t z_{it} + (s_{ij} + p_i) u_{ij} - \bar{d}_i (1 - u_{ij}) \leq \sum_{t=r_j + s_{ij} + 1}^{\bar{d}_j - p_j + 1} t z_{jt} \tag{25}$$

$$i = 0, \ldots, n \quad j = 1, \ldots, n \quad i \neq j$$

$$\sum_{t=0}^{r_j-1} z_{jt} = 0 \qquad j = 1, \ldots, n \tag{26}$$

$$\sum_{t=(\bar{d}_j - p_j + 1) + 1}^{T} z_{jt} = 0 \qquad j = 1, \ldots, n \tag{27}$$

$$z_{00} = 1 \tag{28}$$

The objective (20) is the maximization of the total profit including the tardiness penalties and environmental costs during processing and setup operations. Constraints (21) specify that at each time $t$, the machine can start at most one job. Constraints (22) indicate that an accepted order has at least a successor. Constraints (23) impose to an accepted order $j = 1, \ldots, n$ to have exactly one preceding order. Constraints (24) restrict the starting time of each order to the interval defined from its release date to its deadline minus its processing time. Constraints (25) precise precedence relationship between two orders, guaranteeing that if order $i$ precedes directly order $j$, its starting time must be defined at least at a period after its release date $r_j$ and the setup operation $s_{ij}$, and after $p_i$ the processing of order $i$. Constraints (26) and (27) prevent each order to be processed before its release date and after its deadline. Finally, constraint (28) forces the dummy order to starts the sequence at time 0.

*4.3. Performances*

This subsection provides performances comparison between each of the presented MILP, the ATI formulation [13] and the disjunctive formulation [12]. Models are first compared in term of spatial complexity, and then a comparative analysis is conducted on two benchmarks $B$ and $B'$ of 18 instances each. The benchmark $B$ differs from $B'$ on setup-times values which are all set to 0. The tested instances come from [12] and correspond to instances with $n = 10, 15, 20$, $\tau \in \{0.1, 0.5\}$, and $R \in \{0.1, 0.5, 0.9\}$.

Table 2 displays the number of variables and constraints of each formulation using Landau notation.

**Table 2.** Spatial complexity of each model.

|  | #Variables | #Constraints |
|---|---|---|
| Disjunctive | $\mathcal{O}(n^2) + \mathcal{O}(nT)$ | $\mathcal{O}(n^2) + \mathcal{O}(nT)$ |
| TI On/Off | $\mathcal{O}(n^2) + \mathcal{O}(nT)$ | $\mathcal{O}(n^2T)$ |
| TI Pulse | $\mathcal{O}(n^2) + \mathcal{O}(nT)$ | $\mathcal{O}(n^2)$ |
| ATI | $\mathcal{O}(n^2T)$ | $\mathcal{O}(nT)$ |

As can be seen, ATI formulation is disadvantaged by its number of decision variables. The other formulations have the same number of variables in the worst case. The TI Pulse formulation benefits from fewer constraints than the other formulations with $\mathcal{O}(n^2)$ constraints, coming from the precedence constraints.

Results of the tests on the benchmarks $B$ and $B'$ are reported in Tables 3 and 4 . Each line within the tables corresponds to 6 instances of same $n$ with diverse $\tau$ and $R$ values. The number of feasible and optimal solutions found by the models are reported in columns #fea and #opt respectively. Finally, average solving time in seconds (column $\overline{cpu}$), average CPLEX gap (column $\overline{gap}$), standard deviation for the solving time (column $\sigma_{cpu}$) and standard deviation for the gap (column $\sigma_{gap}$) are presented for each batch of instances. The last line of the tables gives a summary of performances of each model by displaying the total number of feasible and optimal solutions, the average solving time, the average gap and the average standard deviation values across all instances.

**Table 3.** Models performances on benchmark $B$.

|  | Disjunctive | | | | | | TI On/Off | | | | | | TI Pulse | | | | | | ATI | | | | | |
|---|---|---|---|---|---|---|---|---|---|---|---|---|---|---|---|---|---|---|---|---|---|---|---|---|
| $n$ | #fea | #opt | $\overline{cpu}$ | $\overline{gap}$ | $\sigma_{cpu}$ | $\sigma_{gap}$ | #fea | #opt | $\overline{cpu}$ | $\overline{gap}$ | $\sigma_{cpu}$ | $\sigma_{gap}$ | #fea | #opt | $\overline{cpu}$ | $\overline{gap}$ | $\sigma_{cpu}$ | $\sigma_{gap}$ | #fea | #opt | $\overline{cpu}$ | $\overline{gap}$ | $\sigma_{cpu}$ | $\sigma_{gap}$ |
| 10 | 6 | 6 | 52 | 0 | 68 | 5 | 6 | 6 | 0.68 | 0 | 0.5 | 0 | 6 | 6 | 12 | 0 | 27 | 0 | 6 | 6 | 6.04 | 0 | 9 | 0 |
| 15 | 6 | 2 | 2424 | 4.53 | 1822 | 9 | 6 | 6 | 10 | 0 | 8 | 0 | 6 | 5 | 671 | 0.13 | 1450 | 0.3 | 6 | 5 | 764 | 0.002 | 1440 | 0 |
| 20 | 6 | 1 | 3002 | 6 | 1470 | 5 | 6 | 6 | 197 | 0 | 459 | 0 | 6 | 6 | 260 | 0 | 387 | 0 | 6 | 4 | 2046 | 0.01 | 1600 | 0 |
|  | 18 | 9 | 1826 | 3.51 | 1120 | 6.3 | 18 | 18 | 69 | 0 | 155 | 0 | 18 | 17 | 314 | 0.04 | 620 | 0.1 | 18 | 15 | 938 | 0.004 | 1016 | 0 |

**Table 4.** Models performances on benchmark $B'$.

|  | Disjunctive | | | | | | TI On/Off | | | | | | TI Pulse | | | | | | ATI | | | | | |
|---|---|---|---|---|---|---|---|---|---|---|---|---|---|---|---|---|---|---|---|---|---|---|---|---|
| $n$ | #fea | #opt | $\overline{cpu}$ | $\overline{gap}$ | $\sigma_{cpu}$ | $\sigma_{gap}$ | #fea | #opt | $\overline{cpu}$ | $\overline{gap}$ | $\sigma_{cpu}$ | $\sigma_{gap}$ | #fea | #opt | $\overline{cpu}$ | $\overline{gap}$ | $\sigma_{cpu}$ | $\sigma_{gap}$ | #fea | #opt | $\overline{cpu}$ | $\overline{gap}$ | $\sigma_{cpu}$ | $\sigma_{gap}$ |
| 10 | 6 | 6 | 3.17 | 0 | 5 | 0 | 6 | 1 | 3318 | 7 | 690 | 10 | 6 | 6 | 50 | 0 | 37 | 0 | 6 | 6 | 0.7 | 0 | 0.6 | 0 |
| 15 | 6 | 6 | 75 | 0 | 68 | 0 | 6 | 0 | 3600 | 529 | 0 | 320 | 6 | 3 | 2567 | 1 | 1530 | 1.7 | 6 | 5 | 609 | 0.1 | 1465 | 0.6 |
| 20 | 6 | 1 | 3300 | 6 | 750 | 6.3 | 6 | 0 | 3600 | 1504 | 0 | 850 | 4 | 0 | 3600 | 35 | 0 | 52 | 6 | 6 | 449 | 0 | 830 | 0 |
|  | 18 | 13 | 1687 | 2 | 274 | 2.1 | 18 | 1 | 3506 | 680 | 231 | 393 | 16 | 9 | 2072 | 12 | 522 | 18 | 18 | 17 | 529 | 0.03 | 765 | 0.1 |

In Table 3, models are tested on benchmark $B$, which is without setup times. The results show that both TI formulations prevail on the other ones in terms of solving time and solution quality. The TI On/Off is even better than the TI Pulse, finding all optimal solutions of $B$ in a minute on average.

According to Table 4, the TI formulations are rapidly overwhelmed in contrast to the ATI and the disjunctive formulations. The solution quality rapidly decreases proportionally to $n$ for the TI models. In addition, both TI models cannot find some feasible solutions of $B'$. Following the spatial complexity analysis, an explanation can be proposed. The TI formulations are limited by the number of constraints and their nature such as big-M constraints to preserve precedence. Even if the ATI model has a polynomial number of variables, it dominates all the MILP, finding almost all optimal solutions.

## 5. Heuristic Approaches

As the OAS problem in its basic form is NP-hard [9], heuristic solving approaches have been developed. In Section 5.1, the principle of the FR heuristic is presented and implemented for each of the provided formulations. In Section 5.2, a population-based metaheuristic is described and applied to the considered problem.

### 5.1. Fix-and-Relax Heuristics

According to Absi et al. [42], FR heuristic consists in building iteratively a solution from the consecutive solving of relaxed sub-models (or simplified versions) of the studied problem by fixing the value of decision variables deduced from the previously solved sub-problems.

#### 5.1.1. Principle

FR heuristic procedure involves an Observation Window (OW) of length $\sigma_k$ overlapping $\delta_{k+1}$ periods between two successive steps $k$ and $k+1$. In this study, the OW length and the number of overlapping steps remain fixed, thus, for all $k$ $\sigma_k = \sigma$ and $\delta_k = \delta$. Two successive steps of the procedure are illustrated in Figure 2.

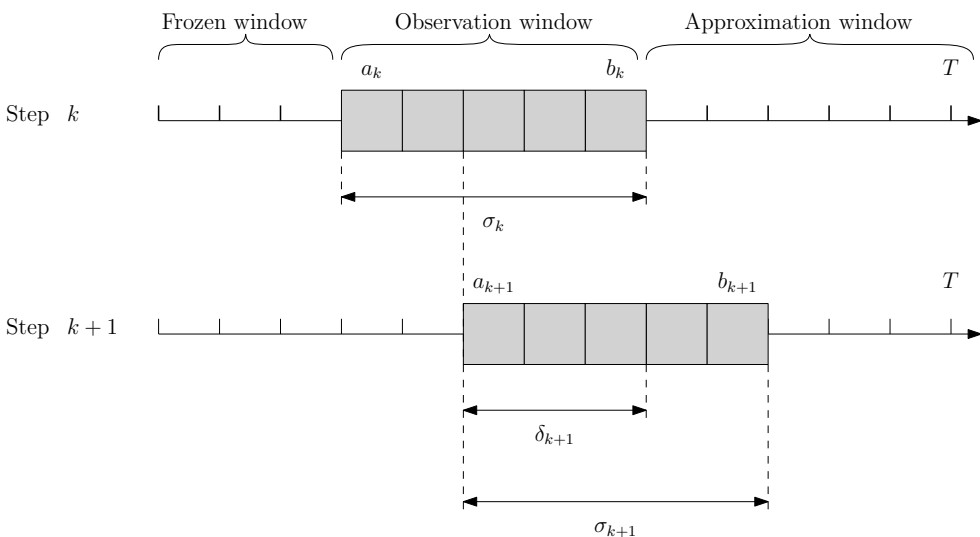

**Figure 2.** FR procedure between steps $k$ and $k+1$. (Reprinted from [56]).

At each step $k$, the decision variables are partitioned into 3 sets according to parameters $a_k$ and $b_k$: a Frozen Window (FW) comprising variables indexed between 0 and $a_k - 1$, an OW in which variables indexes fall between $a_k$ and $b_k$, and an Approximation Window (AW) for variables indexed after $b_k + 1$. At step $k > 1$, the values of the decision variables within the FW are known and integrated into the submodel $Pr_k$. Moreover, constraints containing decision variables within the OW are completely taken into account, whereas constraints involving decision variables in the AW are either dropped or simplified.

Steps of the FR heuristic are described in Algorithm 1. FR heuristic takes as inputs $Pr$ the problem, $\sigma$ the OW length and $\delta$ the number of periods overlapping.

Initial steps from lines 1 to 3 consist in setting $k = 0$ and fixing $a = 0$, $b = \sigma - 1$, while the main loop ends when $b \geq T$. The loop instructions are described from lines 4 to 11. Line 5, the submodel $Pr_k$ is solved. From lines 6 to 8, $k$, $a_k$ and $b_k$ are updated as follows : $k$ is incremented by 1, $a_k$ is set to $b_k - \delta$, and the ending time of the OW $b_k$ is incremented by $\sigma - \delta$ steps. Line 9, the if-statement sets $b_k$ to $T$, preventing $b_k$ to overflow. Finally, at line 13, the last model is solved.

---

**Algorithm 1:** Fix-and-Relax Procedure

---

**Input:** $Pr$: problem; $\sigma$: OW length; $\delta$: overlapping step

1   $k = 0$
2   $a_k = 0$
3   $b_k = \sigma - 1$
4   **while** ($b_k < T$) **do**
5      Solve $Pr_k$
6      $k = k + 1$
7      $a_k = b_k - \delta$
8      $b_k = b_k + \sigma - \delta$
9      **if** ($b_k > T$) **then**
10        $b_k = T$
11     **end**
12   **end**
13   Solve $Pr_k$

---

### 5.1.2. Adaptation

As the developed models are time-indexed, the partitioning of the decision variables follows the time horizon, distinguishing the FW $t = 0, \ldots, a_k - 1$ from the OW $t = a_k, \ldots, b_k$ and the AW $t = b_k + 1, \ldots, T$ where $T$ is the horizon.

As the considered problem includes sequence-dependent setup times, which contribute to the high complexity of the problem, the heuristic approach aims at estimating a simplified version of the problem with constant setup times $\tilde{s}_j$ for orders $j$ that will possibly be produced in the AW. Therefore, the following binary decision variables are introduced in $Pr_k$:

- $\alpha_j$: equals 1 if order $j$ is scheduled between $0, \ldots, b_k$; 0 otherwise;
- $\beta_j$: equals 1 if order $j$ is scheduled between $b_k + 1, \ldots, T$; 0 otherwise

These decision variables split the set of jobs $0, \ldots, n$ into two disjoint subsets $A$ and $B$. Jobs of $A$ are sequenced in the OW while jobs of $B$ are scheduled after $b_k + 1$ (Figure 3).

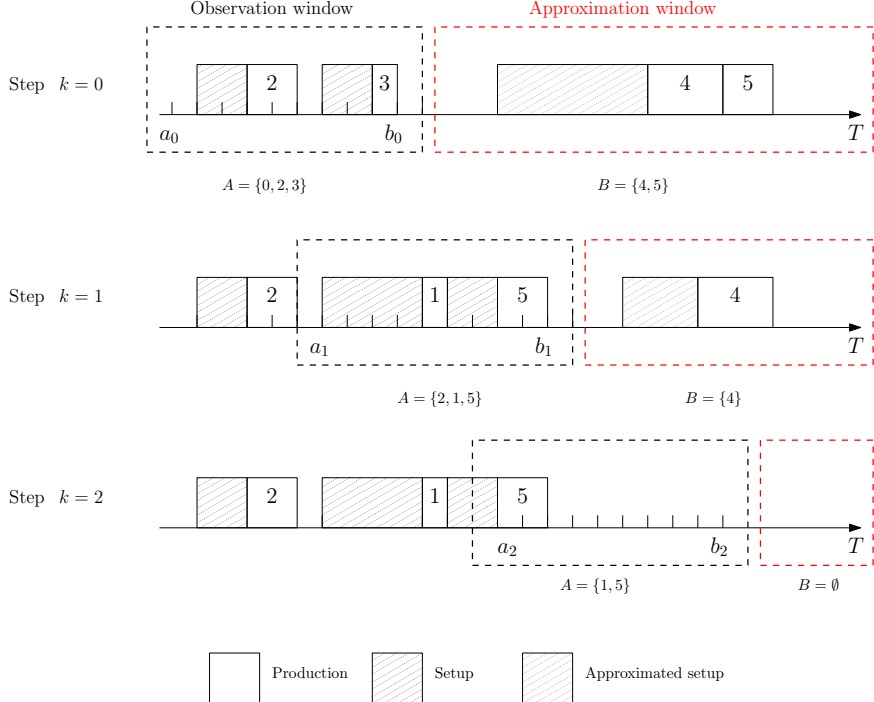

**Figure 3.** Illustration of the developed strategy on an example of $n = 5$ jobs with $\sigma = 9$ and $\delta = 4$.

During the procedure at step $k$, the binary decision variables $x_{jt}$ or $z_{jt}$ such that $t = a_k, \ldots, b_k$ are only considered, which reduce in width the subproblem. This is indicated in Figure 3 when the horizontal axis is graduated. Moreover, the decision variables $x_{jt}$ or $z_{jt}$ associated with already sequenced jobs are evicted from $Pr_k$. This reduces in height the subproblem. The set of non-processed jobs at each step $k$ is denoted as $J_k$. Only the decision variables of the last processed order $l$ in the previous subproblems are kept in $Pr_k$ in order to have the correct setup to make the connection with the next job. Thus, the model shrinks in size iteration by iteration. At each step $k$, the sub-problem model $Pr_k$ integrates the following constraint:

$$\sum_{j \in J_k} \beta_j (\tilde{s}_j + p_j) \leq T - b_k \tag{29}$$

Constraint (29) evaluates that the production of orders, and their setup operations in the AW cannot last more than $T - b_k$ time units. Setup operations of produced orders in the AW have a fixed duration defined by $\tilde{s}_j$. Constraint (29) is, in a way, similar to a time-budget constraint, with an allowance of $T - b_k$ units of production and setup.

### 5.1.3. On/Off Model

The TI On/Off model described in Section 4.1 is adapted to suit to the FR heuristic procedure. The previously introduced decision variables $\alpha_j$ and $\beta_j$ are defined by the group of constraints (30), for all $j \in J_k$.

$$\sum_{t=a_k}^{\min(b_k, \bar{d}_j - p_j)} x_{jt} \leq p_j \alpha_j \qquad j \in J_k \tag{30a}$$

$$\alpha_j + \beta_j \leq 1 \qquad j \in J_k \tag{30b}$$

$$\alpha_l = 1, \quad \beta_l = 0 \tag{30c}$$

Constraints (30a) give value 1 to $\alpha_j$ as soon as order $j$ is processed within the OW, i.e., from $a_k$ to $b_k$, provided that it can be processed completely before its deadline $\bar{d}_j$, and 0 if the order is not produced yet. Constraints (30b) ensure that an order $j$ is either starting before the end of the OW ($\alpha_j = 1$) or completely produced after ($\beta_j = 1$) or is not scheduled at all ($\alpha_j = 0$, $\beta_j = 0$). Constraints (30c) indicate that the previously processed job in the previous step is fixed in the OW and cannot be processed in the AW.

To make precedence constraints (14) and (15) relevant only for orders in the OW, constraints (7) and (8) are modified in (31) and (32), respectively.

$$\sum_{\substack{j \in J_k \\ i \neq j}} u_{ij} \leq \alpha_i \qquad i \in J_k \cup \{l\} \tag{31}$$

$$\sum_{\substack{j \in J_k \cup \{l\} \\ i \neq j}} u_{ji} = \alpha_i \qquad i \in J_k \tag{32}$$

Constraints (31) state that an order $i$ scheduled in the OW can have at most a successor order $j$. Constraints (32) ensure that if order $i$ is processed during the OW, it must have exactly a predecessor order $j$. These reformulated constraints imply that any order $i$ that are scheduled in the AW have $u_{ij}$ values set to zero, as a consequence the right-hand side of constraints (14) and (15) are canceled.

Expressions of constraints (33) and (34) are replaced by

$$\sum_{t'=\max\{r_j,a_k\}}^{t-1} y_{jt'} \geq s_{ij}(u_{ij} + x_{jt} - 1) - s_{ij}\beta_j \tag{33}$$

$$i \in J_k \cup \{l\} \quad j \in J_k$$
$$t = \max\{r_j, a_k\}, \ldots, \min\{\bar{d}_j, b_k\}$$

Constraints (33) ensure that when an order $j$ starts in the AW ($\beta_j = 1$), the right-hand side is canceled and thus the $y_{jt}$ are free. This implies that the setup times between order $i$ and $j$ are not forced to last at least $s_{ij}$ time-units.

$$\sum_{t'=\max\{a_k,r_i\}+1}^{t-1} x_{it'} \geq p_i(u_{ij} + y_{jt} - 1) - p_i\beta_j \tag{34}$$

$$i = J_k \cup \{l\} \quad j \in J_k \quad i \neq j$$
$$t = \max\{r_j, a_k\}, \ldots, \min\{\bar{d}_j, b_k\}$$

In the same way, constraints (34) ensure that when an order $j$ starts in the AW ($\beta_j = 1$), the right-hand side of the constraint is canceled. This means that if an order $i$ (whether $\beta_i = 1$ or $\beta_i = 0$) precedes an order $j$ that is known to be processed in the AW, the setup operation order $j$ is not constrained to starts $p_i$ units after the processing of order $i$. The objective function (2) is rewritten as follows, where $lb = \max\{r_j, a_k\}$ and $ub = \min\{\bar{d}_j, b_k\}$.

$$\textbf{maximize} \sum_{j \in J_k} (\alpha_j e_j - w_j T_j - \sum_{t=lb}^{ub} (x_{jt} + y_{jt})c_{jt} + r\beta_j e_j) \tag{35}$$

In expression (35), the real profit is calculated for any order $j \in J_k$ that is scheduled in the OW. The profit of orders that are scheduled in the AW is approximated by $r\beta_j e_j$ with $r = 0.8$ this is the revenue discounted by a rate $r$. It encourages the model to schedule orders as much as possible in the OW.

### 5.1.4. Pulse Model

With respect to the Pulse model presented in Section 4.2, the formulation of the heuristic strategy is more direct as this model uses decision variables to represent whether an order starts at a specific time period ($z_{jt} = 1$) or not ($z_{jt} = 0$).

In the same manner as the TI On/Off model, the decision variables $\alpha_j$ and $\beta_j$ are incorporated into $Pr_k$, taking into account the characteristics of the TI Pulse formulation. Group of constraints (36) gather the definition of the aforementioned decision variables for all $j \in J_k$.

$$\sum_{t=0}^{b_k} z_{jt} \leq \alpha_j \qquad j \in J_k \tag{36a}$$

$$\alpha_j + \beta_j \leq 1 \qquad j \in J_k \tag{36b}$$

Constraints (36a) give value 1 to $\beta_j$ if order $j$ starts in the OW. Constraints (36b) link $\alpha_j$ and $\beta_j$, and allow order $j$ to starts in the OW ($\alpha_j = 1$) or starts in the AW ($\alpha_j = 1$) or not starting at all. In the OW, precedence between orders must be considered. Therefore, constraints (25) are rewritten for all $i \in J_k \cup \{l\}$ and $j \in J_k$ given that $i \neq j$, where $lb_1 = \max\{r_i, a_k\}$, $lb_2 = \max\{r_j + s_{ij}, a_k\} + 1$, $ub_1 = \min\{b_k - p_i + 1, \bar{d}_i - p_i + 1\}$ and $ub_2 = \min(\bar{d}_j - p_j + 1, b_k - p_j + 1)$.

$$\sum_{t=lb_1}^{ub_1} tz_{it} + (p_i + s_{ij})u_{ij} - \bar{d}_i(1 - u_{ij}) + E_{ij} \leq \sum_{t=lb_2}^{t=ub_2} tz_{jt} \tag{37a}$$

$$i \in J_k \cup \{l\} \quad j \in J_k \quad i \neq j$$

$$E_{ij} = -\bar{d}_i(1 - \alpha_i) - \bar{d}_i(1 - \alpha_j) + \bar{d}_i(\alpha_i - \alpha_j)$$
$$i \in J_k \cup \{l\} \quad j \in J_k \quad i \neq j \tag{37b}$$

Constraints (37a) guarantee that if order $i$ precedes order $j$, the starting time of order $j$ is at least planned after the setup operation $s_{ij}$ and the processing $p_i$ of order $i$. These constraints examine only orders $i$ and $j$ scheduled before the end of the OW defined by $b_k$, provided that they can be processed. To enforce this, the term $E_{ij}$ defined in Equation (37b) aims at canceling the precedence constraint whenever $\alpha_i = 0$. This helps to eliminate precedence between an order $i$ that does not appear in the OW and any other order, thus improving the solving time.

A truth table is presented in Table 5 in order to enumerate each value that $E_{ij}$ takes depending on the values of $\beta_i$ and $\beta_j$.

**Table 5.** Truth table for $E_{ij}$.

| $\beta_i$ | $\beta_j$ | $E_{ij}$ |
|:---:|:---:|:---:|
| 1 | 1 | 0 |
| 1 | 0 | 0 |
| 0 | 1 | $-2d_i$ |
| 0 | 0 | $-2d_i$ |

Finally, the constraints (22) and (23) with respect to $u_{ij}$ decision variables are modified accordingly.

$$\sum_{\substack{j \in J_k \\ i \neq j}} u_{ij} \leq \alpha_i \quad i \in J_k \cup \{l\} \tag{38}$$

$$\sum_{\substack{j \in J_k \cup \{l\} \\ i \neq j}} u_{ji} = \alpha_i \quad i \in J_k \tag{39}$$

Constraints (38) state that an order $i$ starting in the OW should have at least a successor order $j$. Constraints (39) ensure that if order $i$ is scheduled in the OW, it must have exactly a predecessor order $j$. This implies that any order that are outside the OW have $u_{ij}$ values set to zero. The objective function is replaced by expression (40), where $lb = \max\{r_j, a_k\}$, $ub = \min\{\bar{d}_j - p_j + 1, b_k\}$.

$$\textbf{maximize} \sum_{j \in J_k} \left( \sum_{t=lb}^{ub} z_{jt}(f_{jt+p_j-1} - \left( \sum_{t'=0}^{p_j-1} c_{jt+t'} - \sum_{\substack{i=0 \\ i \neq j}}^{n} u_{ij} \sum_{t'=1}^{s_{ij}} c_{jt-t'} \right)) + r\beta_j f_{jb_k} \tag{40}$$

In expression (40), the real profit is calculated for orders $j \in J_k$ which are scheduled in the OW. Otherwise, the profit of orders that are planned in the AW is approximated by $r\beta_j f_{jb_k}$ with $r = 0.8$. That is, the expected profit at the end of the OW with a discounted returns of rate $r$. It prevents the model to delay orders that can be scheduled in the OW.

An overview of three steps of the FR heuristic procedure for the TI Pulse model on an example of $n = 5$ jobs is presented in Figure 4. It illustrates how decision variables $z_{jt}$ are managed in the implementation. A two-entry table is utilized to represent these variables for $j \in J_k$ and $t = 0, \ldots, T$ at each step $k$. The light gray area represents decision variables that are not considered.

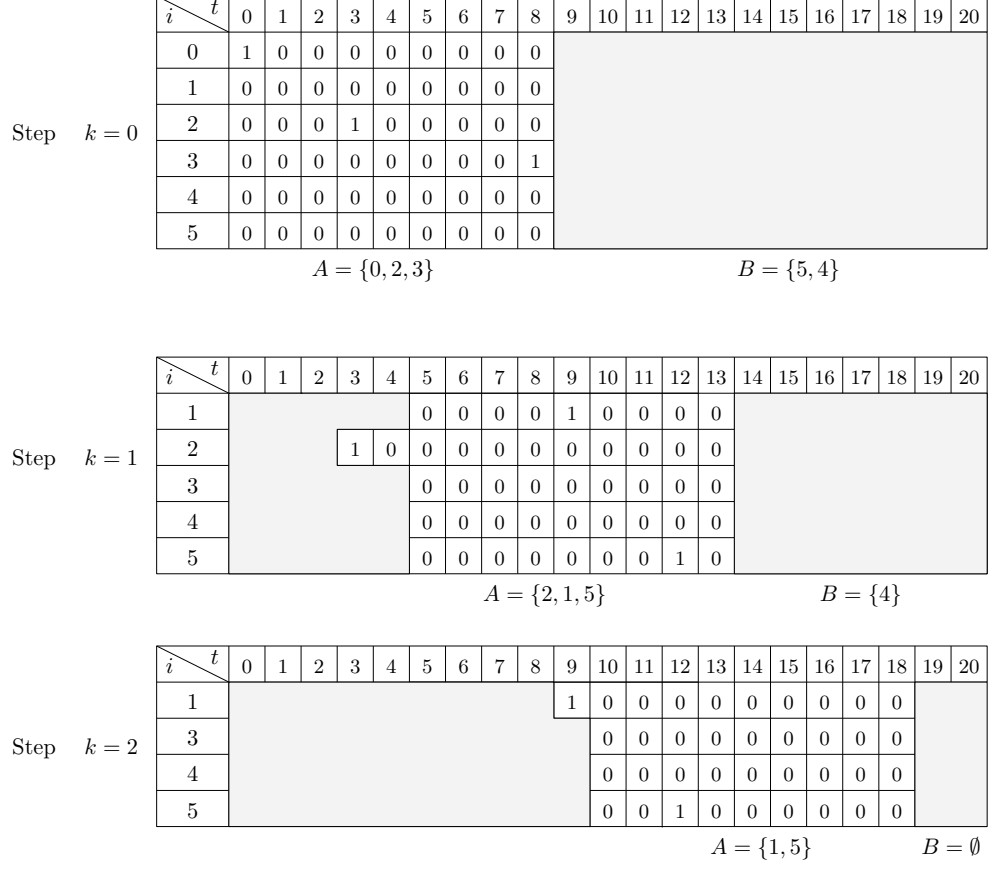

**Figure 4.** FR TI Pulse example with $n = 5$ jobs, $\sigma = 9$ and $\delta = 4$.

### 5.2. Dynamic Island Model

The Dynamic Island Model (DIM) introduced in [14] is applied to the considered problem. DIM is a framework designed as an adaptive operator selection mechanism for an EA in order to solve an optimization problem $(S, f)$. To resume, this technique allows a better exploration and exploitation of the search space $S$ by reinforcing inertia of the best islands (movements) through a stochastic migration process rewarding the most promising island in terms of best improvement according to $f$, while maintaining a certain diversity. In this scheme, the population of each island evolves independently with a classical EA. Table 6 presents a summary of the nomenclature used in this paper.

**Table 6.** Nomenclature.

| Symbol | Description | Domain |
|:---:|:---:|:---:|
| $\nu$ | Number of islands | $\mathbb{N}$ |
| $D$ | Square matrix that stores the feedback from migration | $[0,1]^{\nu \times \nu}$ |
| $M$ | Transition matrix between islands | $[0,1]^{\nu \times \nu}$ |
| $P_i$ | Population of island $i$ | $S^N$ |
| $o_i$ | Mutation operator of island $i$ | $S \to S$ |
| $p_m$ | Mutation rate | $[0,1]$ |
| $\alpha$ | Learning rate | $[0,1]$ |
| $\beta$ | Noise rate | $[0,1]$ |
| $N$ | Initial population size in each island | $\mathbb{N}$ |
| *itmax* | Number of iterations | $\mathbb{N}$ |
| $f$ | Fitness function | $S \to \mathbb{R}$ |
| $p_c$ | Crossover rate | $[0,1]$ |

### 5.2.1. Principle

The general principles of the DIM are presented in Algorithm 2. This algorithm takes as inputs an optimization problem $(S, f)$, values for parameters $\alpha$ and $\beta$, $N$ the number of individual in each island, the number of maximal iterations *itmax*, the mutation rate $p_m$ and the crossover rate $p_c$. The algorithm returns a solution $s$, which is the best solution among the populations at the end of the iterations.

The first instructions from line 1 to 4 consist in initializing $D$ to zero and $M$ to an equiprobable value for each pair $(i, j)$ of islands. Then, line 3 $P_i$ is populated with individuals (greedy, random). In line 4, variable *it* is set to zero. The main while loop is described from line 5 to 21. This loop breaks when the maximum number of iteration is reached.

A for loop, from line 8 to 14, ranging from 1 to $v$ is used to apply a generic steady-state EA to each island. Line 5, two individuals $p_1$ and $p_2$ from $P_i$ are selected in order to provide children from a crossover operator, counting 50% of the population size. Then, children are mutated using the operator $o_i$ with a probability of $p_m$. Finally, line 9, the top 50% worst solutions from $P_i$ are picked out and replaced by *children* solutions.

Following the for loop, line 17, the transition matrix $M$ is updated according to $D$, rewarding only the best islands. Then, line 18, a migration process is performed to the whole population $P$ with the transition matrix $M$. Finally, in line 19, the data matrix $D$ is updated in order to store the impact of the transition to the solution quality. The counter *it* is incremented at the end of the while loop in line 20. The final instruction line 22 consists in getting the best solution $s$ among the populations. This solution $s$ is returned by the algorithm, line 23. The next subsections detail precisely update, migrate and analyze procedures used in Algorithm 2.

---

**Algorithm 2:** DIM Algorithm

**Input:** $(S, f)$ : optimization problem; $\alpha$ : learning rate ; $\beta$ : noise rate; $N$ : initial population size in each island; *itmax* : number of iterations; $p_c$ : crossover rate; $p_m$ : mutation rate

**Result:** $s$ : a solution

1   $D_{ij} = 0$;
2   $M_{ij} = \frac{1}{v}$ ;
3   $P_i = \text{populate}(N)$ ;
4   $it = 0$ ;
5   **while** $(it < itmax)$ **do**
6     **for** $(i = 1, i \leq v, i = i + 1)$ **do**
7       $r = \text{genRandom}(0,1)$ ;
8       **if** $(r < p_c)$ **then**
9         $p_1, p_2 = \text{select}(P_i)$ ;
10        *children* = $\text{crossover}(p_1, p_2)$ ;
11        $o_i(children, p_m)$ ;
12        $P_i = P_i \cup children$;
13        *worst* = $\text{selectWorst}(f, P_i)$ ;
14        $\text{replace}(P_i, worst, children)$ ;
15       **end**
16     **end**
17     $\text{update}(M, D)$ ;
18     $\text{migrate}(P, M)$;
19     $\text{analyze}(D, f)$;
20     $it = it + 1$ ;
21   **end**
22   $s = \text{best}(f, \cup_{i=1}^{v} P_i)$ ;
23   **return** $s$

---

### Update

This process allows modifying the transitions $M_{ij}$ only for the best islands $i$. This is possible with intermediates vectors: a reward vector $R$ and a stochastic noise vector $N$ ($\sum_{j=1}^{\nu} N_j = 1$) defined for each island $j = 1, \ldots, \nu$. A set $B$ is also defined to store the indexes of the best island according to $D$.

$$R_j = \begin{cases} \frac{1}{B} & \text{if } j \in B \\ 0 & \text{otherwise.} \end{cases}$$

$$B = \underset{i=1,\ldots,\nu}{\mathrm{argmax}} \, D_{ij} \tag{41}$$

The transition $M_{ij}$ for each pair of islands $(i, j)$ is then updated as follows.

$$M_{ij} = (1 - \beta)(\alpha M_{ij} + (1 - \alpha)R_j) + \beta N_j \tag{42}$$

As stated in [14], only the islands where the individuals obtained significant improvements benefit from a reward, thus reinforcing the best operators. In the calculation of $M_{ij}$, the parameter $\alpha$ represents the inertia (or exploitation) and $\beta$, the amount of noise necessary to explore other search space areas.

### Migrate

The migration process involves every individual from each island and the transition matrix $M$. In this process, a random number $r \sim \mathcal{U}(0, 1)$ is drawn from a uniform distribution for each individual of the population $P_i$. For each destination island $j$, if $r < M_{ij}$ then the individual is sent to island $j$ and remove from island $i$.

### Analyze

This process retrieves for each pair of islands $(i, j)$ the best fitness among individuals that have migrated from island $i$ to island $j$ in the previous iteration, thus measuring the benefits of the transition $i \rightarrow j$.

$$D_{ij} = \max_{s \in P_{ji}} f(s)$$

$$P_{ji} = \{ s \in P_j s \text{ comes from } i \} \tag{43}$$

The 'analyze' process needs the originating island of individuals at previous iteration. Consequently, this information is stored in the solution.

In Figure 5, both the transition matrix $M$ and the population size of each island are represented with a directed graph (DG). Each edge $(i, j)$ from vertex $i$ to vertex $j$ has a weight that corresponds to $M_{ij}$. As the edges represent stochastic transition between vertices, the sum of the weights of the out-coming edges must be 1. Finally, the vertices in this diagram are proportional to the size of the population.

#### 5.2.2. Solution Representation

A solution $s$ is represented by a sequence $\{i_1, \ldots, i_k, \ldots i_\kappa\}$ of size $\kappa \leq n$ where $i_k = 1, \ldots, n$ is the $k$th order to be scheduled. Moreover, an integer $origin = 1, \ldots, \nu$ is used to represent the residence island at previous iteration. Associated completion times, $C_{i_k}$, for all $k = 1, \ldots, \kappa$ are computed in the decoding phase (Figure 6).

Completion times $C_{i_k}$ are calculated so that each order $i_k$ starts as soon as possible at time period $ST_{i_k} = \max(C_{i_{k-1}}, r_{i_k})$, thus $C_{i_k} = ST_{i_k} + p_{i_k} + s_{i_{k-1}i_k}$ when $C_{i_k} > \bar{d}_{i_k}$ the order is rejected.

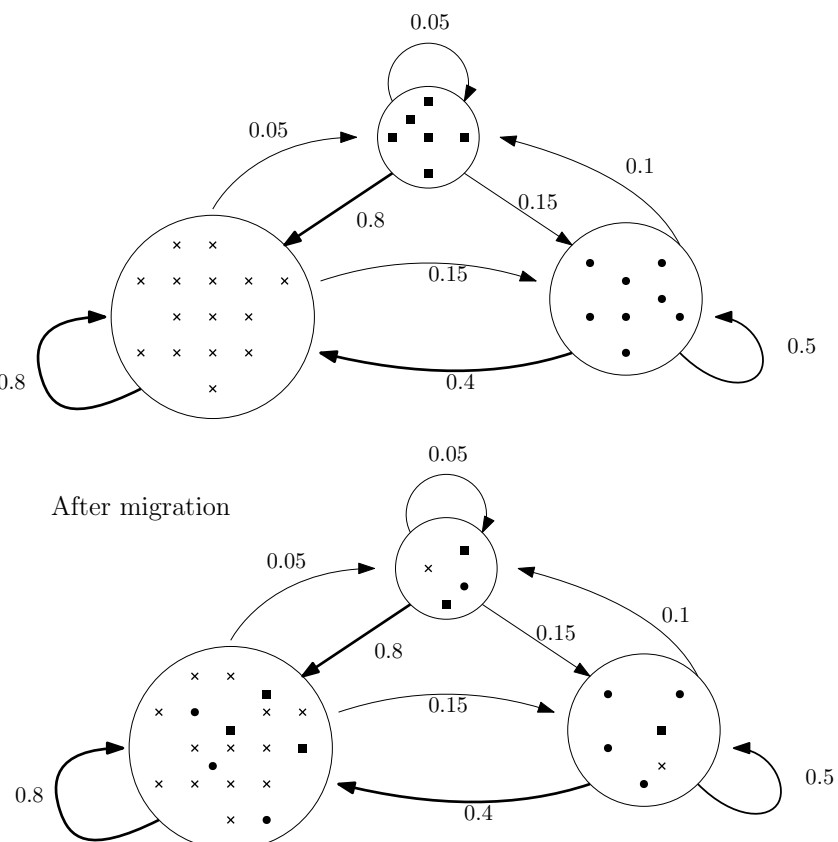

**Figure 5.** Directed Graph (DG) for an example with 3 islands.

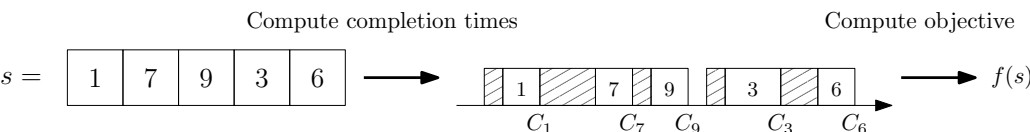

**Figure 6.** Decoding a solution by updating completion times.

### 5.2.3. Mutations Operators

Each island is characterized by a mutation operator. The majority of these operators can lead to the rejection of orders in the sequence. Completion times are thus updated accordingly whenever an operator is applied.

#### Add

This operator picks randomly a order $\bar{a}$ among the rejected order list and inserts it at a random position $k$ in the sequence.

#### Swap

The operator swap takes randomly two orders $i_j$ and $i_k$ and swaps their position $j$ and $k$ in the sequence.

#### Exchange

This operator picks randomly a rejected order $\bar{a}$ and an order $i_k$ in the sequence and replaces them. That is, the order $i_k$ is rejected and the order $\bar{a}$ is put at position $k$ in the sequence.

#### Shift

This operator aims at shifting all the orders at the best starting time in terms of energy cost without causing tardiness or rejection. This operator examines for all order $i_k$ the best

insertion period by computing exhaustively the energy cost between the starting times $ST_A$ and $ST_B$ and choosing $t_{best}$ where the energy cost is minimal.

$$
\begin{aligned}
t_{best} &= \underset{t=ST_A,\ldots,ST_B}{\operatorname{argmin}} \sum_{t'=t}^{t+p_{i_k}+s_{i_{k-1}i_k}} c_{i_k t'} \\
ST_A &= \max(C_{i_{k-1}}, r_{i_k}) \\
ST_B &= \min(ST_{i_{k+1}}, d_{i_k}) - p_{i_k} - s_{i_{k-1}i_k}
\end{aligned}
\tag{44}
$$

Scramble

The scramble operator picks two position $k$ and $j$ randomly between $1,\ldots,\kappa$ and shuffles the sequence between these positions.

Inversion

This operator generates randomly two positions $k$ and $j$ between $1,\ldots,\kappa$ with $k < j$. Then, between $k$ and $j$ the sequence is inverted.

For the purpose of efficiency, a version of the operators *Add* and *Exchange* that take into consideration the revenue load ratio ($RL_j = \frac{e_j}{p_j}$) has been developed. For the operator *Add*, this consists in systematically add the best rejected order according to this ratio. For the *Exchange* operator, this implies choosing the worst order according to $RL$ and replace it the best rejected order, in accordance with the revenue load ratio. These versions are respectively referred as *AddGreedy* and *ExchangeGreedy*.

### 5.2.4. Crossover Operator

For the considered problem, the crossover operator from in [57] has been utilized. This crossover takes as input two parents solutions $p_1$ and $p_2$. The sequence of $p_1$ is examined at each position. For each position, orders are successively add up to the children sequence, using a random number to either inserts the order from $p_1$ or the one from $p_2$. To prevent duplicate orders in the child solution, the orders appearing in the children that have already been inserted are not considered (Figure 7).

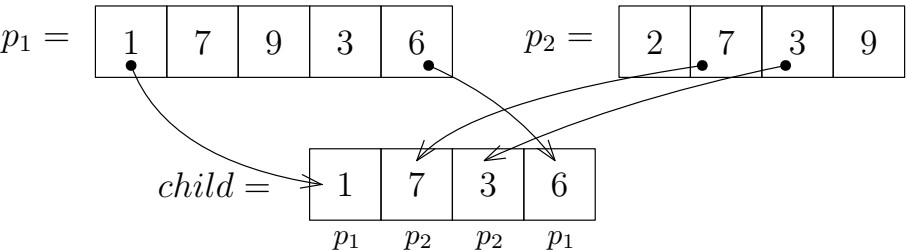

**Figure 7.** Example of the crossover operator application.

### 5.2.5. Initial Population

The initial population is built with 80% of random solution. The other 20% of the population are created with greedy heuristics.

Random Solution

This procedure starts by inserting all orders to the sequence and shuffle it. Then, completion times are calculated.

Earliest Released Job

This greedy heuristic sorts the sequence of orders using the Earliest Released Job rule. Completion times are computed accordingly.

m-ATCS

This greedy heuristic is taken from the work of Cesaret et al. [38]. The procedure starts with $L = 1, \ldots, n$ the list of orders, $l = 0$ the initial scheduled order and $t = 0$ the starting time. It iteratively inserts order $i$ from $L$ into a sequence $s$ using the Apparent Tardiness Cost with Setups (ATCS) index at time $t$, knowing the previous scheduled order $l$. The formula is given in Equation (45). The order $i$ with the largest ATCS($i,l,t$) is added in $s$ and erased from $L$, then $t$ is set to $\max(t, r_i) + p_i + s_{l,i}$. This continues until $L$ is empty. Completion times are computed according to this sequence.

$$\text{ATCS}(i, l, t) = \frac{e_i}{p_i} e^{\max(d_i - p_i - t, 0) / \bar{p}} e^{-s_{l,i} / \bar{s}} \tag{45}$$

In (45), $\bar{s}$ corresponds to the average setup times and $\bar{p}$ refers to the average processing times.

## 6. Computational Results

### 6.1. Experimental Design

Experimental designs aim at determining levels of influence and interaction of external factors on a process. In this research, both the FR heuristics and the metaheuristic depend on many factors such as the OW length or the setup values for the FR heuristics and such as the mutation, crossover or learning rates for the DIM metaheuristic.

Taguchi method has been chosen to carry out these experimental designs for the proposed approaches. The reasons are that this easy-to-implement method has proven itself to be efficient and robust to tune GA and heuristics in this domain. Details are available in the Appendix A.

### 6.2. Benchmark and Material

The benchmark is from Chen et al. [12] with sequence-dependent setup times. It contains 45 instances with various number of orders $n = 10, 15, 25, 50, 100$ and two parameters to control the tardiness factor $\tau = 0.1, 0.5, 0.9$ and the due date range $R = 0.1, 0.5, 0.9$. These parameters aim at having a diverse set of instances.

Figures 8 and 9 illustrate the impact of $\tau$ and $R$ on instance characteristics. The instances represented share the same processing times $p$ but differ on release dates $r$, due dates $d$ and deadlines $\bar{d}$ depending on the value of $\tau$. In the figures, the possible production span between $r_j$ and $d_j$ for an order $j$ is indicated as a white rectangle, whereas the penalty interval between $d_j$ and $\bar{d}_j$ is indicated as a gray rectangle.

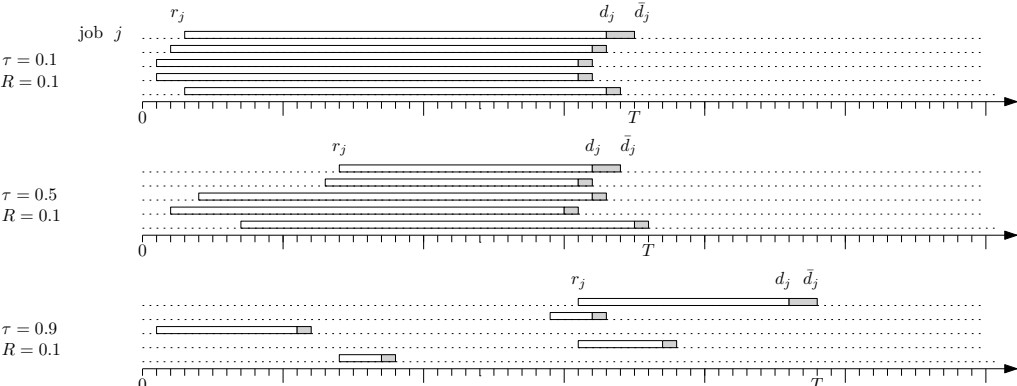

**Figure 8.** Influence of $\tau$ on instance characteristics (with $R$ fixed).

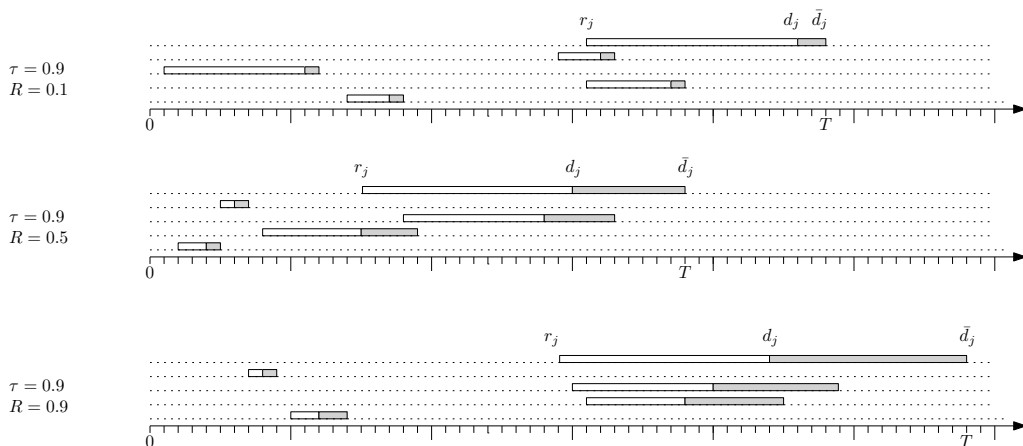

**Figure 9.** Influence of *R* on instance characteristics (with $\tau$ fixed).

As displayed in Figure 8 , the value of $\tau$ determines the dispersion of the release dates ($r_j$) as well as the production slack, that is, the length of the production interval between release and due dates ($d_j$). The greater $\tau$ is, the more scattered the release dates are and the less flexible the production is. Note that more flexibility increases the combinatorial aspect of the problem.

In Figure 9, the impact of *R* on the instances is shown. *R* controls the length of the interval for each order *j* between due dates ($d_j$) and deadlines ($\bar{d}_j$), that is, the penalty interval. *R* affects also the slope of the tardiness penalty ($w_j$). When *R* is large, the penalty interval is large as well and the tardiness penalty decreases less quickly.

The implementation of the FR heuristics has been made in C++17 using the IBM CPLEX library *v.*12.10. Moreover, the DIM metaheuristic has been developed in C++17. The tests have been performed on a desktop computer with Intel i5 2GHz CPU processor and 4GB RAM. Solving time is limited to 3600 s for each instance.

According to the Taguchi analysis performed on the FR TI Pulse, the best setting is the following : $\sigma = 3 \times p_{\max}$, $\delta = 50\%$, $\tilde{s}_j$ = Max. With regard to the TI On/Off model, the most performant setting is $\sigma = 2 \times p_{\max}$, $\delta = 25\%$, $\tilde{s}_j$= Max. For the FR heuristics, the solving time of the subproblem at each step has been limited to $nT/3600$ s. For the DIM, the best tuning is the following: $popsize = 100$, $\alpha = 0.8$, $\beta = 0.1$, $itmax = 1000$, $p_c = 0.9$ and $p_m = 0.7$.

*6.3. Results*

First, results of the FR heuristics TI Pulse and TI On/Off are presented in comparison with the best known solution (BKS) found (either ATI or Chen et al. [12], which is shown in Table A8 in the Appendix A), then with the most efficient FR heuristic and the best and the average performance of the DIM among 10 runs. In Tables 7 and 8, each line represents an instance with its setting $n, \tau, R$ and $T$. Objective value, CPU time (s) and deviation from the BKS (%) are given for each approach. The last line presents the average performances across the benchmark.

**Table 7.** FR Heuristics vs. BKS.

| | | | | | FR TI Pulse | | | FR TI On/Off | | |
|---|---|---|---|---|---|---|---|---|---|---|
| *n* | *τ* | *R* | *T* | BKS | Obj | CPU (s) | Dev (%) | Obj | CPU (s) | Dev (%) |
| 10 | 0.1 | 0.1 | 117 | 118.71 | 110.70 | 40 | 7 | 80.66 | 0.5 | 32 |
| | | 0.5 | 145 | 107.51 | 96.50 | 16 | 10 | 63.62 | 0.3 | 41 |
| | | 0.9 | 156 | 93.62 | 70.66 | 8 | 25 | 62.65 | 0.3 | 33 |
| | 0.5 | 0.1 | 96 | 98.54 | 78.63 | 11 | 20 | 55.74 | 0.3 | 43 |
| | | 0.5 | 141 | 98.62 | 90.49 | 5 | 8 | 50.72 | 0.3 | 49 |
| | | 0.9 | 139 | 102.47 | 97.45 | 2 | 5 | 63.64 | 0.5 | 38 |
| | 0.9 | 0.1 | 121 | 57.70 | 54.73 | 0 | 5 | 19.91 | 0.2 | 65 |
| | | 0.5 | 147 | 75.34 | 67.19 | 0 | 11 | 38.79 | 0.1 | 49 |
| | | 0.9 | 133 | 106.51 | 91.57 | 1 | 14 | 63.33 | 0.3 | 41 |
| 15 | 0.1 | 0.1 | 148 | 135.48 | 124.50 | 96 | 8 | 101.60 | 2 | 25 |
| | | 0.5 | 217 | 212.58 | 193.22 | 91 | 9 | 158.47 | 1.5 | 25 |
| | | 0.9 | 217 | 160.09 | 128.78 | 63 | 20 | 115.28 | 2 | 28 |
| | 0.5 | 0.1 | 194 | 147.32 | 119.42 | 31 | 19 | 77.54 | 0.9 | 47 |
| | | 0.5 | 163 | 160.25 | 141.02 | 34 | 12 | 102.41 | 1 | 36 |
| | | 0.9 | 251 | 135.75 | 110.26 | 7 | 19 | 62.65 | 1 | 54 |
| | 0.9 | 0.1 | 184 | 117.44 | 107.52 | 1 | 8 | 68.19 | 0.5 | 42 |
| | | 0.5 | 199 | 138.58 | 127.12 | 1 | 8 | 106.94 | 0.6 | 23 |
| | | 0.9 | 175 | 90.64 | 76.66 | 1 | 15 | 48.52 | 0.6 | 46 |
| 25 | 0.1 | 0.1 | 279 | 305.01 | 248.22 | 235 | 19 | 236.26 | 6 | 23 |
| | | 0.5 | 315 | 241.99 | 199.24 | 177 | 18 | 177.38 | 6 | 27 |
| | | 0.9 | 403 | 283.09 | 221.98 | 149 | 22 | 198.40 | 4 | 30 |
| | 0.5 | 0.1 | 301 | 274.12 | 240.26 | 95 | 12 | 200.54 | 3 | 27 |
| | | 0.5 | 362 | 248.69 | 199.72 | 115 | 20 | 152.16 | 2 | 39 |
| | | 0.9 | 312 | 278.56 | 203.87 | 122 | 27 | 194.93 | 3 | 30 |
| | 0.9 | 0.1 | 305 | 207.05 | 185.94 | 2 | 10 | 129.26 | 1 | 38 |
| | | 0.5 | 309 | 231.91 | 216.08 | 3 | 7 | 165.36 | 1 | 29 |
| | | 0.9 | 424 | 262.14 | 235.25 | 3 | 10 | 162.52 | 1 | 38 |
| 50 | 0.1 | 0.1 | 522 | 500.15 | 456.23 | 486 | 9 | 441.30 | 49 | 12 |
| | | 0.5 | 688 | 513.91 | 525.74 | 475 | −2 | 447.01 | 35 | 13 |
| | | 0.9 | 754 | 456.25 | 493.81 | 426 | −8 | 476.80 | 39 | −5 |
| | 0.5 | 0.1 | 614 | 419.09 | 426.14 | 369 | −2 | 359.54 | 16 | 14 |
| | | 0.5 | 636 | 552.56 | 449.62 | 331 | 19 | 377.07 | 20 | 32 |
| | | 0.9 | 766 | 541.41 | 428.06 | 305 | 21 | 405.90 | 12 | 25 |
| | 0.9 | 0.1 | 583 | 384.72 | 348.98 | 13 | 9 | 213.86 | 5 | 44 |
| | | 0.5 | 701 | 448.39 | 410.96 | 18 | 8 | 267.95 | 5 | 40 |
| | | 0.9 | 752 | 502.56 | 406.21 | 76 | 19 | 292.97 | 11 | 42 |
| 100 | 0.1 | 0.1 | 1211 | 635.78 | 893.60 | 1168 | −41 | 859.96 | 256 | −35 |
| | | 0.5 | 1402 | 639.23 | 972.02 | 1080 | −52 | 812.77 | 191 | −27 |
| | | 0.9 | 1486 | 595.37 | 938.90 | 1026 | −58 | 830.60 | 416 | −40 |
| | 0.5 | 0.1 | 1033 | 529.71 | 1010.09 | 829 | −91 | 946.93 | 70 | −79 |
| | | 0.5 | 1288 | 502.53 | 868.34 | 789 | −73 | 798.09 | 51 | −59 |
| | | 0.9 | 1401 | 505.32 | 803.64 | 809 | −59 | 785.44 | 39 | −55 |
| | 0.9 | 0.1 | 1111 | 852.49 | 863.02 | 354 | −1 | 692.78 | 14 | 19 |
| | | 0.5 | 1186 | 877.93 | 783.00 | 280 | 11 | 574.79 | 15 | 35 |
| | | 0.9 | 1191 | 899.47 | 767.20 | 298 | 15 | 579.24 | 12 | 36 |
| | Avg. | | | | | 232 | 2 | | 29 | 22 |

**Table 8.** FR Heuristic TI Pulse, DIM best and average vs. BKS.

| *n* | *τ* | *R* | *T* | BKS | FR TI Pulse | | | DIM Best | | | DIM Avg | | |
|---|---|---|---|---|---|---|---|---|---|---|---|---|---|
| | | | | | Obj | CPU (s) | Dev (%) | Obj | CPU (s) | Dev (%) | Obj | CPU (s) | Dev (%) |
| 10 | 0.1 | 0.1 | 117 | 118.71 | 110.70 | 40 | 7 | 118.71 | 9 | 0 | 118.70 | 13 | 0 |
| | | 0.5 | 145 | 107.51 | 96.50 | 16 | 10 | 107.51 | 17 | 0 | 107.51 | 19 | 0 |
| | | 0.9 | 156 | 93.62 | 70.66 | 8 | 25 | 93.62 | 6 | 0 | 93.62 | 13 | 0 |
| | 0.5 | 0.1 | 96 | 98.54 | 78.63 | 11 | 20 | 98.54 | 3 | 0 | 98.54 | 7 | 0 |
| | | 0.5 | 141 | 98.62 | 90.49 | 5 | 8 | 98.62 | 4 | 0 | 98.62 | 7 | 0 |
| | | 0.9 | 139 | 102.47 | 97.45 | 2 | 5 | 102.47 | 5 | 0 | 102.47 | 9 | 0 |
| | 0.9 | 0.1 | 121 | 57.70 | 54.73 | 0 | 5 | 57.70 | 0 | 0 | 57.70 | 0 | 0 |
| | | 0.5 | 147 | 75.34 | 67.19 | 0 | 11 | 75.34 | 0 | 0 | 75.34 | 0 | 0 |
| | | 0.9 | 133 | 106.51 | 91.57 | 1 | 14 | 106.51 | 1 | 0 | 106.51 | 3 | 0 |
| 15 | 0.1 | 0.1 | 148 | 135.48 | 124.50 | 96 | 8 | 135.48 | 14 | 0 | 135.47 | 17 | 0 |
| | | 0.5 | 217 | 212.58 | 193.22 | 91 | 9 | 212.58 | 22 | 0 | 210.39 | 23 | 1 |
| | | 0.9 | 217 | 160.09 | 128.78 | 63 | 20 | 160.09 | 22 | 0 | 159.85 | 23 | 0 |
| | 0.5 | 0.1 | 194 | 147.32 | 119.42 | 31 | 19 | 147.32 | 13 | 0 | 147.32 | 18 | 0 |
| | | 0.5 | 163 | 160.25 | 141.02 | 34 | 12 | 160.25 | 14 | 0 | 160.24 | 18 | 0 |
| | | 0.9 | 251 | 135.75 | 110.26 | 7 | 19 | 135.75 | 15 | 0 | 135.75 | 19 | 0 |
| | 0.9 | 0.1 | 184 | 117.44 | 107.52 | 1 | 8 | 117.45 | 1 | 0 | 117.44 | 1 | 0 |
| | | 0.5 | 199 | 138.58 | 127.12 | 1 | 8 | 138.58 | 2 | 0 | 138.58 | 4 | 0 |
| | | 0.9 | 175 | 90.64 | 76.66 | 1 | 15 | 90.64 | 1 | 0 | 90.64 | 3 | 0 |
| 25 | 0.1 | 0.1 | 279 | 305.01 | 248.22 | 235 | 19 | 300.98 | 26 | 1 | 298.48 | 31 | 2 |
| | | 0.5 | 315 | 241.99 | 199.24 | 177 | 18 | 241.97 | 28 | 0 | 240.60 | 29 | 1 |
| | | 0.9 | 403 | 283.09 | 221.98 | 149 | 22 | 281.86 | 26 | 0 | 280.14 | 28 | 1 |
| | 0.5 | 0.1 | 301 | 274.12 | 240.26 | 95 | 12 | 274.11 | 25 | 0 | 271.71 | 26 | 1 |
| | | 0.5 | 362 | 248.69 | 199.72 | 115 | 20 | 248.69 | 26 | 0 | 245.09 | 27 | 1 |
| | | 0.9 | 312 | 278.56 | 203.87 | 122 | 27 | 277.46 | 23 | 0 | 275.35 | 26 | 1 |
| | 0.9 | 0.1 | 305 | 207.05 | 185.94 | 2 | 10 | 207.05 | 11 | 0 | 207.05 | 15 | 0 |
| | | 0.5 | 309 | 231.91 | 216.08 | 3 | 7 | 231.91 | 12 | 0 | 231.91 | 16 | 0 |
| | | 0.9 | 424 | 262.14 | 235.25 | 3 | 10 | 262.14 | 9 | 0 | 262.14 | 19 | 0 |
| 50 | 0.1 | 0.1 | 522 | 500.15 | 456.23 | 486 | 9 | 519.13 | 37 | −4 | 516.86 | 38 | −3 |
| | | 0.5 | 688 | 513.91 | 525.74 | 475 | −2 | 588.41 | 40 | −14 | 583.04 | 42 | −13 |
| | | 0.9 | 754 | 456.25 | 493.81 | 426 | −8 | 555.61 | 38 | −22 | 554.74 | 41 | −22 |
| | 0.5 | 0.1 | 614 | 419.09 | 426.14 | 369 | −2 | 512.85 | 37 | −22 | 509.35 | 38 | −22 |
| | | 0.5 | 636 | 552.56 | 449.62 | 331 | 19 | 558.91 | 37 | −1 | 550.35 | 39 | 0 |
| | | 0.9 | 766 | 541.41 | 428.06 | 305 | 21 | 558.96 | 37 | −3 | 555.22 | 38 | −3 |
| | 0.9 | 0.1 | 583 | 384.72 | 348.98 | 13 | 9 | 384.69 | 27 | 0 | 381.57 | 31 | 1 |
| | | 0.5 | 701 | 448.39 | 410.96 | 18 | 8 | 445.34 | 24 | 1 | 440.77 | 30 | 2 |
| | | 0.9 | 752 | 502.56 | 406.21 | 76 | 19 | 487.83 | 27 | 3 | 480.44 | 33 | 4 |
| 100 | 0.1 | 0.1 | 1211 | 635.78 | 893.60 | 1168 | −41 | 1043.23 | 58 | −64 | 1040.51 | 61 | −64 |
| | | 0.5 | 1402 | 639.23 | 972.02 | 1080 | −52 | 1144.50 | 69 | −79 | 1139.56 | 70 | −78 |
| | | 0.9 | 1486 | 595.37 | 938.90 | 1026 | −58 | 1149.95 | 67 | −93 | 1142.88 | 72 | −92 |
| | 0.5 | 0.1 | 1033 | 529.71 | 1010.09 | 829 | −91 | 1175.30 | 61 | −122 | 1165.29 | 62 | −120 |
| | | 0.5 | 1288 | 502.53 | 868.34 | 789 | −73 | 1008.30 | 62 | −101 | 1002.20 | 64 | −99 |
| | | 0.9 | 1401 | 505.32 | 803.64 | 809 | −59 | 1039.50 | 60 | −106 | 1025.40 | 63 | −103 |
| | 0.9 | 0.1 | 1111 | 852.49 | 863.02 | 354 | −1 | 992.65 | 56 | −16 | 984.71 | 58 | −16 |
| | | 0.5 | 1186 | 877.93 | 783.00 | 280 | 11 | 814.19 | 43 | 7 | 799.54 | 48 | 9 |
| | | 0.9 | 1191 | 899.47 | 767.20 | 298 | 15 | 829.48 | 43 | 8 | 797.70 | 48 | 11 |
| | | Avg. | | | | 232 | 2 | | 26 | −14 | | 29 | −13 |

Table 7 displays a comparison between the BKS and the two FR heuristics. The FR Pulse heuristic obtains 2% deviation on average from the best solutions found by the models in [12,13], with reasonable solving time. For small to medium instances, the heuristic finds solutions with 13% deviation on average. For large instances from $n = 50$ to 100, the heuristic finds better solutions than the exact approaches (10/45). This is particularly true on instances with $\tau = 0.1$ and $R = 0.1$. The FR Pulse heuristic is able to improve up to 91% of the objective. The FR TI On/Off heuristic has 22% deviation from the BKS with an average solving time of 30 s. On average, the solution founds are of lower quality for this heuristic compared to the TI Pulse. However, this heuristic can find solutions with less computational effort (10 times less). The FR TI On/Off finds better solutions than the exact approaches for 7 instances with $n = 50, 100$ and specific $\tau$ and $R$ values.

FR heuristics are better for small values of $\tau$ on average. This is probably because the FR heuristic is guided by the accumulation of information in the OW to make selecting and sequencing decisions. When $\tau = 0.1$, orders are available from the beginning, therefore, the local choices (in the OW) better match the global choice and thus the optimal solution. If $\tau$ is large, orders are available at different points of the horizon making the decision in the OW much more local. Consequently, the heuristic will possibly be led to make a succession of bad choices, without being able to backtrack. When $\sigma$ is larger, FR heuristic benefits from more information to make decisions but it increases the computational effort.

Table 8 displays a comparison between the BKS, the FR TI Pulse heuristic, and the best and average solution found by DIM. The DIM metaheuristic (best) obtains $-14\%$ deviation on average from the BKS for a resolution time of 26 s on average. The tests allowed to find 13 better solutions compared to the MILPs. The robustness of the metaheuristic on small instances (0.2%) is as good as on large instances (0.8%). Finally, the performance in terms of deviation from the BKS is $-35\%$ for large instances on average and 0.08% for small to medium instances on average.

In terms of deviation, the DIM metaheuristic dominates all the approaches. However, we notice that the performances depend on the characteristics of the instances. For $\tau = 0.9$, the DIM metaheuristic is less robust and need much more iterations to find the optimal solutions. This can be explained by the non-specificity of the mutation operators to tackle instances with scattered release dates and less flexibility. When $n \geq 50$, the FR heuristics become better than MILPs with reasonable solving times. Moreover, a clear improvement is noticeable with instances for which $\tau = 0.1$.

## 7. Conclusions and Perspectives

This paper proposes two new mathematical formulations for a rather recent research, that is, the OAS problem with release dates, sequence-dependent setup times, TOU costs and $CO_2$ emissions periods. The provided MILPs are time-indexed; however, these new exact models are limited to solve medium and large instances in presence of sequence-dependent setup times. Without setup, the TI On/Off formulation is the most competitive.

In this context, original FR heuristics that approximate setup are developed, taking advantage of time indexation. Better solutions have been found by these heuristics. According to the results obtained on the state-of-the-art benchmark, the best version of FR heuristic is the one with the TI Pulse formulation. Moreover, in this paper, a population-based metaheuristic is also developed. The latter is based on Dynamic Island Model framework. This procedure can solve small to large instances within half a minute on a personal computer with average performance features.

Future work can be dedicated to the extension of the problem to other systems such as parallel machines or floor shop. Determining mathematical properties of the studied problem can be undertaken. In addition, further analysis can be devoted to the instance settings $\tau$ and $R$ and TOU tariffs policy. Moreover, an extensive analysis on other $CO_2$ emissions reduction policies is an interesting prospect, as this work only focuses on the taxes on carbon emissions. For instance, a limitation on carbon emissions could be incorporated in the formulation.

Extended tests on larger instances $n > 100$ shall be performed in order to assess the performance of FR heuristics. For instance, Benders decomposition approaches can be developed on the presented time-indexed formulations and compare the performances with the provided FR heuristics. As for the DIM, more specific mutation operators shall be developed in order to tackle instances that are difficult to solve. Other solution representation can also be explored in order to compare it with the sequence-based representation.

**Author Contributions:** Conceptualization, M.B., O.M. and A.Y.; methodology, M.B., O.M. and A.Y.; software, M.B.; validation, O.M. and A.Y.; investigation, M.B., O.M. and A.Y.; writing—original draft preparation, M.B.; writing—review and editing, O.M. and A.Y. ; supervision, O.M. and A.Y.; funding acquisition, O.M. and A.Y. All authors have read and agreed to the published version of the manuscript.

**Funding:** This research was funded by the Grand-Est Region and the Aube Department in France. The authors gratefully acknowledge this support.

**Institutional Review Board Statement:** Not applicable.

**Informed Consent Statement:** Not applicable.

**Data Availability Statement:** The data are available in the following Github repository https://github.com/worldstar/OpenGA/tree/master/instances/SingleMachineOASWithTOU (accessed on 14 January 2020).

**Acknowledgments:** We gratefully acknowledge and express our appreciation to the Editor and the anonymous reviewers who provided feedback.

**Conflicts of Interest:** The authors declare no conflicts of interest.

## Abbreviations

The following abbreviations are used in this manuscript:

| | |
|---|---|
| OAS | Order Acceptance Scheduling |
| TOU | Time-of-Use |
| FR | Fix-and-Relax |
| DIM | Dynamic Island Model |
| GA | Genetic Algorithm |
| EA | Evolutionary Algorithm |
| GHG | Greenhouse Gas |
| MILP | Mixed Integer Linear Program |
| ATI | Arc-Time indexed |
| TI | Time-Indexed |
| TEC | Total Energy Cost Consumption |
| FW | Frozen Window |
| AW | Approximation Window |
| OW | Observation Window |
| BKS | Best Known Solution |

## Appendix A

*Appendix A.1. Experimental Design*

The Taguchi method is a fractional factorial design of experiments developed by Dr. Taguchi in the late 1940s [58]. It limits the number of parameter settings to be tested to estimate the optimal one, when testing all of them would take a tremendous amount of time. To put things into perspective, when five parameters taking two values, a brute-force approach would consist in testing $2^5$ settings on a sample of 15 instances. A high estimate of 3600 s per instance gives a total of 1.7 million seconds to carry out the tests, which is not reasonable. To avoid such thing, Taguchi designs orthogonal arrays, which consider a reduced amount of settings to run. The results of each experiment are converted to a

Signal-to-Noise (S/N) ratio in order to estimate the effects of control factors on the data mean and variation. The S/N ratio response table provides the best tuning for the process.

Using Minitab 19 [59], the procedure involves the following steps:

1. Identify factors
2. Characterize levels of factors
3. Select an orthogonal array
4. Run experiments and collect the response data
5. Analyze the experimental data
6. Identify the optimal levels of factors
7. Validate experiment

First, control factors with their range of values are selected. A large catalog of designs are available, as required. For instance, with 4 factors and 2 levels by factor, the proposed design referrers as $2^4$ or $L_8$, which means a $P = 8$ runs design. The resulting orthogonal array must be judiciously selected, particularly if factors are dependent. In step 3, the default orthogonal array given by Minitab is chosen. During step 4, experiments are carried out with the proper settings described by the orthogonal array. In each experiment, objective values are collected. As the problem aims at determining factor levels that result in the largest means, the S/N ratio of each experiment $e = 1, \ldots, P$ is calculated with the *Larger-is-Better* formula as follows, with $y_i$ the $i$th objective value and $M$ the sample size:

$$\mathrm{S/N}(e) = -10\log\frac{1}{M}\sum_{i=1}^{M}\frac{1}{y_i^2} \qquad e = 1, \ldots, P \tag{A1}$$

Adjustment have been made to the collected response values in order to properly use the formula described in (A1) and remove statistical biases. First, response data have been smoothed using min-max normalization. Second, as the S/N function takes only strictly positive values, normalized values have been scaled into the interval $[1, 2]$.

*Appendix A.2. Fr Heuristics*

For the tests, two values for the setup times are used, as shown in Table A1.

**Table A1.** Setup times values.

|  | Value for $\tilde{s}_j$ |
|---|---|
| optimistic | $\displaystyle\min_{i\in J_k} s_{ij}$ |
| conservative | $\displaystyle\max_{i\in J_k} s_{ij}$ |

The FR heuristic depends on parameters such as the decision window size $\sigma$ and the number of overlapping periods $\delta$. The impact of these parameters is studied with $\sigma = 2p_{\max}$ and $\sigma = 3p_{\max}$ ($p_{\max} = \max_{j=1,\ldots,n} p_j$) and two values of $\delta$ representing 25% and 50% of the length of the decision window $\sigma$.

For the FR TI Pulse and the TI On/Off formulations, 3 factors and 2 levels are chosen in the design of experiment. Table A2 presents factors A, B and C and their corresponding description and levels (set of possible values).

**Table A2.** Factors and their levels for the FR heuristics.

| Factor | Name | First Level | Second Level |
|---|---|---|---|
| A | Decision window size | $2 \times p_{\max}$ | $3 \times p_{\max}$ |
| B | Overlapping steps | 25% | 50% |
| C | Setup-times approximation | Min | Max |

The orthogonal array for the FR TI formulations is given in Table A3. Each line corresponds to an experiment with its associated settings. For example, in experiment 3, factors A and C have their values fixed at second level and factors B at first level.

**Table A3.** Orthogonal array $L_8$ for the FR heuristics.

| Exp # | A | B | C |
|-------|---|---|---|
| 1 | 1 | 1 | 1 |
| 2 | 1 | 2 | 2 |
| 3 | 2 | 1 | 2 |
| 4 | 2 | 2 | 2 |

Table A4 displays the S/N ratios for the FR Pulse with the Larger-is-Better formula.

**Table A4.** Response table for S/N ratios for the TI FR Pulse.

| Level | $\sigma$ | $\delta$ | $\tilde{s}$ |
|-------|----------|----------|-------------|
| 1 | 3.099 | 2.518 | 3.046 |
| 2 | 2.247 | 2.829 | 2.300 |
| Delta | 0.852 | 0.311 | 0.746 |
| Rank | 1 | 3 | 2 |

Table A5 displays the S/N ratios for the FR On/Off with the Larger-is-Better formula.

**Table A5.** Response table for S/N ratios for the TI FR On/Off.

| Level | $\sigma$ | $\delta$ | $\tilde{s}$ |
|-------|----------|----------|-------------|
| 1 | 2.618 | 2.751 | 2.906 |
| 2 | 3.484 | 3.351 | 3.195 |
| Delta | 0.866 | 0.600 | 0.289 |
| Rank | 1 | 2 | 3 |

*Appendix A.3. Dynamic Island Model*

Table A6 presents the factors and their levels for the metaheuristic. The response table for S/N ratios for the DIM metaheuristic is displayed in Table A7.

**Table A6.** Factors and their levels for the DIM metaheuristic.

| Factor | Name | First Level | Second Level |
|--------|------|-------------|--------------|
| A | *popsize* | 50 | 100 |
| B | $\alpha$ | 0.6 | 0.8 |
| C | $\beta$ | 0.01 | 0.1 |
| D | *itmax* | 500 | 1000 |
| E | $p_c$ | 0.7 | 0.9 |
| F | $p_m$ | 0.3 | 0.7 |

**Table A7.** Response table for S/N ratios for the DIM.

| Level | *popsize* | *itmax* | $\alpha$ | $\beta$ | $p_m$ | $p_c$ |
|-------|-----------|---------|----------|---------|-------|-------|
| 1 | 3.540 | 3.495 | 3.431 | 3.435 | 3.145 | 3.811 |
| 2 | 4.623 | 4.668 | 4.732 | 4.728 | 5.018 | 4.351 |
| Delta | 1.083 | 1.173 | 1.301 | 1.293 | 1.873 | 0.540 |
| Rank | 5 | 4 | 2 | 3 | 1 | 6 |

**Table A8.** Results of the ATI formulation [13] and the disjunctive model of Chen et al. [12].

| | | | | ATI | | | Disjunctive | | | BKS |
|---|---|---|---|---|---|---|---|---|---|---|
| *n* | *τ* | *R* | *T* | Obj | CPU (s) | Gap (%) | Obj | CPU (s) | Gap (%) | |
| 10 | 0.1 | 0.1 | 117 | 118.71 | 1 | 0 | 118.71 | 97 | 0 | 118.71 |
| | | 0.5 | 145 | 107.51 | 1 | 0 | 107.51 | 226 | 0 | 107.51 |
| | | 0.9 | 156 | 93.62 | 1 | 0 | 93.62 | 10 | 0 | 93.62 |
| | 0.5 | 0.1 | 96 | 98.54 | 0 | 0 | 98.54 | 4 | 0 | 98.54 |
| | | 0.5 | 141 | 98.62 | 0 | 0 | 98.62 | 5 | 0 | 98.62 |
| | | 0.9 | 139 | 102.47 | 1 | 0 | 102.47 | 3 | 0 | 102.47 |
| | 0.9 | 0.1 | 121 | 57.70 | 0 | 0 | 57.70 | 0 | 0 | 57.70 |
| | | 0.5 | 147 | 75.34 | 0 | 0 | 75.34 | 0 | 0 | 75.34 |
| | | 0.9 | 133 | 106.51 | 0 | 0 | 106.51 | 0 | 0 | 106.51 |
| 15 | 0.1 | 0.1 | 148 | 135.48 | 49 | 0 | 135.48 | 3600 | 4 | 135.48 |
| | | 0.5 | 217 | 210.15 | 3600 | 2 | 212.58 | 38 | 0 | 212.58 |
| | | 0.9 | 217 | 160.09 | 14 | 0 | 160.09 | 63 | 0 | 160.09 |
| | 0.5 | 0.1 | 194 | 147.32 | 3 | 0 | 147.32 | 420 | 0 | 147.32 |
| | | 0.5 | 163 | 160.25 | 3 | 0 | 160.25 | 2015 | 0 | 160.25 |
| | | 0.9 | 251 | 135.75 | 3 | 0 | 135.75 | 99 | 0 | 135.75 |
| | 0.9 | 0.1 | 184 | 117.44 | 1 | 0 | 117.44 | 0 | 0 | 117.44 |
| | | 0.5 | 199 | 138.58 | 1 | 0 | 138.58 | 0 | 0 | 138.58 |
| | | 0.9 | 175 | 90.64 | 1 | 0 | 90.64 | 0 | 0 | 90.64 |
| 25 | 0.1 | 0.1 | 279 | 305.01 | 192 | 0 | 297.04 | 3600 | 8 | 305.01 |
| | | 0.5 | 315 | 241.99 | 3600 | 1 | 241.01 | 3600 | 1 | 241.99 |
| | | 0.9 | 403 | 283.09 | 385 | 0 | 278.14 | 3600 | 3 | 283.09 |
| | 0.5 | 0.1 | 301 | 274.12 | 135 | 0 | 257.23 | 3600 | 14 | 274.12 |
| | | 0.5 | 362 | 248.69 | 16 | 0 | 239.77 | 3600 | 10 | 248.69 |
| | | 0.9 | 312 | 278.56 | 9 | 0 | 278.56 | 3600 | 2 | 278.56 |
| | 0.9 | 0.1 | 305 | 207.05 | 2 | 0 | 207.05 | 1 | 0 | 207.05 |
| | | 0.5 | 309 | 231.91 | 3 | 0 | 231.91 | 26 | 0 | 231.91 |
| | | 0.9 | 424 | 262.14 | 4 | 0 | 262.14 | 21 | 0 | 262.14 |
| 50 | 0.1 | 0.1 | 522 | 400.88 | 3600 | 32 | 500.15 | 3600 | 9 | 500.15 |
| | | 0.5 | 688 | 416.29 | 3600 | 42 | 513.91 | 3600 | 15 | 513.91 |
| | | 0.9 | 754 | 403.10 | 3600 | 38 | 456.25 | 3600 | 22 | 456.25 |
| | 0.5 | 0.1 | 614 | 419.09 | 3600 | 24 | 406.29 | 3600 | 33 | 419.09 |
| | | 0.5 | 636 | 213.45 | 3600 | 167 | 552.56 | 3600 | 6 | 552.56 |
| | | 0.9 | 766 | 393.34 | 3600 | 50 | 541.41 | 3600 | 11 | 541.41 |
| | 0.9 | 0.1 | 583 | 384.72 | 24 | 0 | 378.81 | 3600 | 15 | 384.72 |
| | | 0.5 | 701 | 448.39 | 45 | 0 | 442.37 | 3600 | 15 | 448.39 |
| | | 0.9 | 752 | 502.56 | 85 | 0 | 475.18 | 3600 | 14 | 502.56 |
| 100 | 0.1 | 0.1 | 1211 | 0.00 | 3600 | - | 635.78 | 3600 | 74 | 635.78 |
| | | 0.5 | 1402 | 0.00 | 3600 | - | 639.23 | 3600 | 83 | 639.23 |
| | | 0.9 | 1486 | 0.00 | 3600 | - | 595.37 | 3600 | 95 | 595.37 |
| | 0.5 | 0.1 | 1033 | 529.71 | 3600 | 130 | 194.98 | 3600 | 557 | 529.71 |
| | | 0.5 | 1288 | 502.53 | 3600 | 116 | 391.17 | 3600 | 172 | 502.53 |
| | | 0.9 | 1401 | 426.45 | 3600 | 158 | 505.32 | 3600 | 121 | 505.32 |
| | 0.9 | 0.1 | 1111 | 488.60 | 3600 | 37 | 852.49 | 3600 | 28 | 852.49 |
| | | 0.5 | 1186 | 877.93 | 3600 | 0 | 618.29 | 3600 | 61 | 877.93 |
| | | 0.9 | 1191 | 899.47 | 3600 | 0 | 747.61 | 3600 | 41 | 899.47 |
| | Avg. | | | | 1382 | 19 | | 2069 | 31 | |

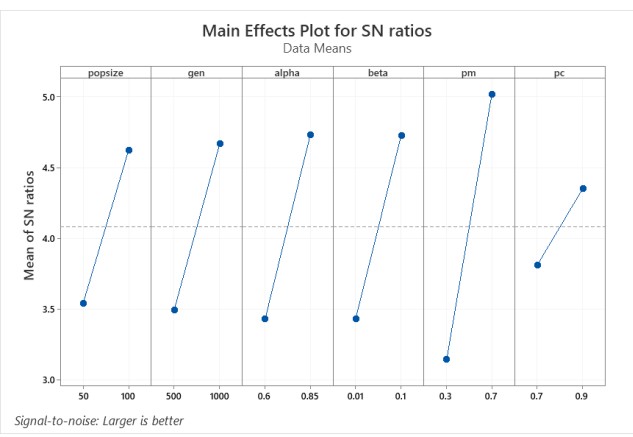

**Figure A1.** S/N ratio for the DIM.

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
