# Peer review of "Exact Methods and Heuristics for Order Acceptance Scheduling Problem under Time-of-Use Costs and Carbon Emissions"

_applsci, doi:10.3390/app11198919_

Round 1
Reviewer 1 Report
ID: applsci-1367560-peer-review-v1
Title: Exact and approximate methods for the Order Acceptance Scheduling problem under time-of-use costs and carbon emissions
The focus of the paper is on a shop scheduling (single machine) problem under order acceptance assumption (OAS), time-of-use (TOU) tariffs and taxed carbon emission periods. The objective is to maximize total profit minus tardiness penalties and environmental costs.
In general, the paper has a good quality. The topic can be interest of both shop scheduling (job shop, flow shop, …) and energy researchers. However, I have some comments about organization and content of the paper. Please see the following comments:
- My main concern is about the methodology of the paper in Section 3. It is fine to assume that TOU tariffs are time dependent and then define interval k = 1, . . . , m for it. However, I cannot understand why CO2 emission should also be time dependent considering interval l = 1, . . . , h. As I know, CO2 emission depends on the amount of fuel or electricity used to complete a task, and so it is time independent (not a matter when the electricity is used, the job still emits same CO2). As such, authors should correct it in the problem definition and model (1)-(19), or they should explain in detail why they have this assumption.
- Another issue is that both “Tax” and “Cap and trade” are introduced in the paper. They are common in the context of energy efficiency and emission reduction. First, I would like to know which one is the main concern of the paper and why? This is a kind of questionable in the current version of the paper. Second, I think authors should explain if their model is extendable to more possible cases?
- “Single machine scheduling” should be added to keywords in Page 1. This helps readers to know that the paper is in the domain of shop scheduling.
- Fig. 2 has never addressed in the body of paper. It is confusing.
- The literature review section lacks discussion and summary. Please add them in more detail.
- The title of most papers in the reference list are lowercase, whereas few are Capitalized each Word (e.g., references 10, 24 and 34). This is a case of inconsistency. Please make all of them lowercase for the sake of simplicity.
- In the literature review, there should be a discussion about shop scheduling (e.g. job shop and flow shop) considering CO2 emissions. I suggest briefly citing the following papers about this issue. [a] Solving the energy-efficient job shop scheduling problem: a multi-objective genetic algorithm with enhanced local search, Journal of Cleaner Production, vol.112, pp. 3361-3375 [b] The impact of various carbon reduction policies on green flowshop scheduling, Applied Energy, vol.249, pp. 300-315
- A common mistake in the paper is about the use of singular and plural nouns:
Page 1: the Fix-and-Relax (FR) heuristics --> the Fix-and-Relax (FR) heuristic
Page 2: introducing it as a criteria --> introducing it as a criterion
Page 15: as an adaptive operator selection mechanisms --> as an adaptive operator selection mechanism
Page 16: for each pair of island --> for each pair of islands
- Other errors:
Page 5: is developped in --> is developed in
Page 9: reported in tables 3 and 4 --> reported in Tables 3 and 4
Page 9: In table 3, models are --> In Table 3, models are
Page 9: According to table 4 --> According to Table 4
…
Page 20: As displayed in figure 8 --> As displayed in Fig. 8
Reviewer 2 Report
The paper is interesting and well structured. Nevertheless, I suggest to the author a major revision taking into account the following comments:
- In the literature approximation methods are methods that have convergence properties or methods such that the gap can be measured in analytic ways. Since the authors are using heuristics I suggest removing the term approximation and replace it with heuristic in order to be more clear.
- I would appreciate a more precise definition of the problem already in the introduction, in order to understand the problem the reader must wait until Section 3. Furthermore, in the introduction, I would appreciate a list of real problems which can be considered the one selected here.
- Since one of the goals of your paper is to "proposes a competitive and robust metaheuristic to fill the gap in the literature" I would appreciate a small section in the literature review related to the newest application of metaheuristic to scheduling problems. See e.g. Machine learning and optimization for production rescheduling in Industry 4.0 Li, Y., Carabelli, S., Fadda, E., Manerba, D., Tadei, R., Terzo, O. International Journal of Advanced Manufacturing Technology, 2020, 110(9-10), pp. 2445–2463 and Reinforcement Learning Algorithms for Online Single-Machine Scheduling Li, Y., Fadda, E., Manerba, D., Tadei, R., Terzo, O. Proceedings of the 2020 Federated Conference on Computer Science and Information Systems, FedCSIS 2020, 2020, pp. 277–283, 9222933 and the references therein.
- in the graph I would explain better than w_j is the slope of the decay.
- Table 1 may divide between variables and parameters
- line 195 TI is not defined
- line 208 "TI On/off" instead of Off? Check all notation in the paper.
- the authors state that the problem is " sequence-dependent setup-times" but the model (2) - (19) does not seem to consider sequence-dependent setup-times. In fact, the setup variables just depend on the job and by the time.
- Constraint (3) why not <= a_j instead of <=1?
- line 240 "Finally fo each"
- The pulse formulation is not introduced properly. Is it just another equivalent model for the same problem?
- In section 4.3 for ease of reading, you can report the link to the instance of paper [9] and [10].
- In Tables 3 and 4, you report #fea and #opt, what kind of information you can extract from these numbers? How is computed the average gap?
- In Tables 3 and 4 the standard deviation of the values is missing.
- After figure 5 there is a dot.
- Table 7 I suggest adding the gap instead of the Objective value.
- The problem that you have to consider is considering just one machine, what about more machines?
- Can you please share online the code that you have developed as well as provide the link to the instances that you have used?
Round 2
Reviewer 1 Report
I have read the paper once more, mostly with the focus on the highlighted sentences. I can see a significant improvement.
The CO2 scheduling challenges are well explained now and the contribution of the submission is also clear. So, the paper can be accepted as it is.
Reviewer 2 Report
You carefully take into account the review. I think that the paper is now ready for publication.